# Stage-resolved Hi-C analyses reveal meiotic chromosome organizational features influencing homolog alignment

Wu Zuo[1,2,8], Guangming Chen[3,4,5,8], Zhimei Gao[3,4,8], Shuai Li[3,4], Yanyan Chen[3,4], Chenhui Huang 🔘 [3,4], Juan Chen[3,4], Zhengjun Chen[1], Ming Lei 🔘 [3,4,6,7 ✉] & Qian Bian 🔘 [3,4 ✉]

During meiosis, chromosomes exhibit dramatic changes in morphology and intranuclear positioning. How these changes influence homolog pairing, alignment, and recombination remain elusive. Using Hi-C, we systematically mapped 3D genome architecture throughout all meiotic prophase substages during mouse spermatogenesis. Our data uncover two major chromosome organizational features varying along the chromosome axis during early meiotic prophase, when homolog alignment occurs. First, transcriptionally active and inactive genomic regions form alternating domains consisting of shorter and longer chromatin loops, respectively. Second, the force-transmitting LINC complex promotes the alignment of ends of different chromosomes over a range of up to 20% of chromosome length. Both features correlate with the pattern of homolog interactions and the distribution of recombination events. Collectively, our data reveal the influences of transcription and force on meiotic chromosome structure and suggest chromosome organization may provide an infrastructure for the modulation of meiotic recombination in higher eukaryotes.

[1] State Key Laboratory of Molecular Biology, Shanghai Institute of Biochemistry and Cell Biology, Center for Excellence in Molecular Cell Science, Chinese Academy of Sciences, 200031 Shanghai, China. [2] University of Chinese Academy of Sciences, 100049 Beijing, China. [3] Ninth People's Hospital, Shanghai Jiao Tong University School of Medicine, 200125 Shanghai, China. [4] Shanghai Institute of Precision Medicine, 200125 Shanghai, China. [5] Key Laboratory of Vector Biology and Pathogen Control of Zhejiang Province, Huzhou University, 313000 Huzhou, China. [6] State Key Laboratory of Oncogenes and Related Genes, Shanghai Jiao Tong University School of Medicine, 200025 Shanghai, China. [7] Key Laboratory of Cell Differentiation and Apoptosis of Chinese Ministry of Education, Shanghai Jiao Tong University School of Medicine, 200025 Shanghai, China. [8]These authors contributed equally: Wu Zuo, Guangming Chen, Zhimei Gao. ✉email: leim@shsmu.edu.cn; qianbian@shsmu.edu.cn

Meiosis is the fundamental process that reduces chromosome numbers by half during sexual reproduction[1–3]. In this process, large segments of chromosomes are exchanged between homologous chromosomes, thereby generating genetic diversity. The meiotic program involves a series of precisely orchestrated chromosome events that couple homologous recombination with dynamic chromosome reorganization[4,5]. During the leptotene stage of the extended meiotic prophase I, programmed DNA double-strand breaks (DSBs) occur on chromosomes, initiating a searching process that promotes the pairing of homologous chromosomes[6–8]. Upon homolog juxtaposition in the zygotene stage, DSBs are converted into recombination intermediates[9]. Concomitantly, a proteinaceous structure called synaptonemal complex (SC) assembles between homolog pairs to stabilize the inter-homolog interactions[10]. During the pachytene stage, SC is fully established, and a fraction of recombination intermediates mature into crossovers (COs)[11]. SC is subsequently disassembled during the diplotene stage, followed by the reductional segregation of homologs in meiosis I and equational segregation of sister chromatids in meiosis II (MII) to generate haploid gametes.

To support the highly specialized meiotic program, meiotic chromosomes adopt a unique conformation comprising linearly arranged loops along chromosome axes[4,12]. The loop array conformation, distinct from the chromatin organization in either interphase or mitosis, allows transcription to take place within the loops stemming from chromosome axes while achieving considerable axial compaction[13]. Meanwhile, DNA sequences near the bases of loops can interact with SC components, as well as recombination machinery, forming a compartment for homologous recombination to occur[14,15]. The sex-dependent differences in chromatin loop sizes and axis lengths correlate with the sexually dimorphic CO frequency[16,17], suggesting that the meiotic chromosome organization may contribute to the global modulation of recombination events. However, whether and how the local variations of meiotic loop array organization across the genome influence homolog interactions and recombination frequency have not been fully understood.

In addition to the reorganization within individual chromosomes, meiotic chromosomes also undergo dramatic telomere-led movements during early meiotic prophase I[18–20], highlighted by a transient "bouquet" conformation at the early zygotene stage, during which all chromosomes coalesce[21]. Two key mechanisms underlie the dynamic repositioning of telomeres. Telomeres are anchored to the inner nuclear membrane (INM) via the interaction between telomere-associated protein TRF1 and the INM-associated TTM complex consisting of TERB1, TERB2, and MAJIN[22–25]. Upon re-localizing to the nuclear periphery, telomeres also associate with the linker of the cytoskeleton and nucleoskeleton (LINC) complex, which transduces the force generated from the cytoskeleton to chromosomes to promote rapid chromosome movement[26–28]. Both mechanisms are essential for homolog pairing and meiosis progression. Of particular note, it has been documented that COs preferentially occur near the telomere during spermatogenesis[29,30]. Whether such a telomere-proximity-dependent distribution of COs reflects underlying variations in chromatin loop organization or differential distribution of recombination factors remains to be elucidated.

Recent advances in the chromosome conformation capture (Hi-C) technique have enabled the characterization of meiotic chromosome organization at a higher resolution. Hi-C studies on mouse and rhesus monkey spermatocytes revealed substantial changes in higher-order chromosome structure features during meiosis[31–35]. The topologically associating domains (TADs) were largely lost by the pachytene stage. On the other hand, the transcriptionally active A and inactive B compartments, though attenuated, were still present during zygotene and pachytene stages[32,34]. While these studies uncovered general principles of meiotic chromosome folding, the roles of these chromosome organization features in regulating the meiotic events remain to be elucidated.

Hi-C characterization of meiotic chromosome organization has also been performed in synchronized *Saccharomyces cerevisiae* populations[36,37], which provided a unique system for examining the temporal progression of meiotic chromosome loops. However, considerable differences exist between yeast and mammalian meiotic chromosome structures. The chromatin loop sizes in yeast are significantly smaller compared to mammals (20 kb in yeast versus hundreds of kb in mammals)[4]. Besides, chromatin loops are anchored at fixed locations along meiotic chromosome axes in yeast, implying less plasticity in the regulation of chromatin loop sizes[37,38]. Furthermore, unlike mammalian genomes, the yeast genome does not contain large domains of transcriptionally active and inactive regions and lacks structural features such as TADs and A/B compartments[39]. Hi-C characterization in mammalian meiocytes at a greater temporal resolution, in a fashion similar to yeast, would improve the understanding of the meiotic chromosome reorganization processes in mammals.

In this study, we employed mouse spermatogenesis as a model system to investigate mammalian meiotic chromosome organization and its functional significance, particularly in the regulation of homolog interactions. By combining spermatogenesis synchronization and Hi-C, we systematically characterized genome architecture throughout all meiotic prophase I substages from preleptotene to diplotene. Our Hi-C analyses unveil two types of large-scale chromosome organizational features that vary along the chromosome axis during early meiotic prophase I, when homolog alignment occurs. First, transcriptionally active and inactive regions form alternating domains consisting of shorter and longer chromatin loops, respectively. Second, the force-transmitting LINC complex promotes the alignment of chromosome ends over a range of up to 20% of chromosome length. The position-dependent variations in both types of organizational features correlate extensively with the extent of homolog alignment and the distribution of recombination events. Taken together, our stage-resolved Hi-C maps reveal the influences of transcriptional activities and LINC-mediated force transmission on meiotic chromosome organization, suggesting that the structural properties of meiotic chromosomes may contribute to the modulation of meiotic recombination during mammalian gametogenesis.

## Results

**Mapping genome architecture throughout mouse spermatogenesis at a high temporal resolution.** To comprehensively characterize chromosome organization changes during mammalian meiosis, we isolated a continuum of cell populations representing different pre-meiotic and meiotic stages throughout mouse spermatogenesis. Using fluorescence-activated cell sorting (FACS) with stringent gating strategies, we isolated the somatic Sertoli cells, the rapidly proliferating spermatogonia cells, and the preleptotene cells from the testes of 7–11-day-old male C57BL/6 mice (Fig. 1a, b), the zygotene-stage spermatocytes from 2-week-old mice (Fig. 1a, c), and the pachytene and diplotene-stage spermatocytes, as well as the MII spermatocytes from 3- to 5-week-old mice (Fig. 1a, d). We also purified the highly transitive leptotene-stage spermatocytes from 16-day-old mice testes using a spermatogenesis synchronization strategy that involves treating young mice 2 days post parturition (dpp) with WIN 18,446 and

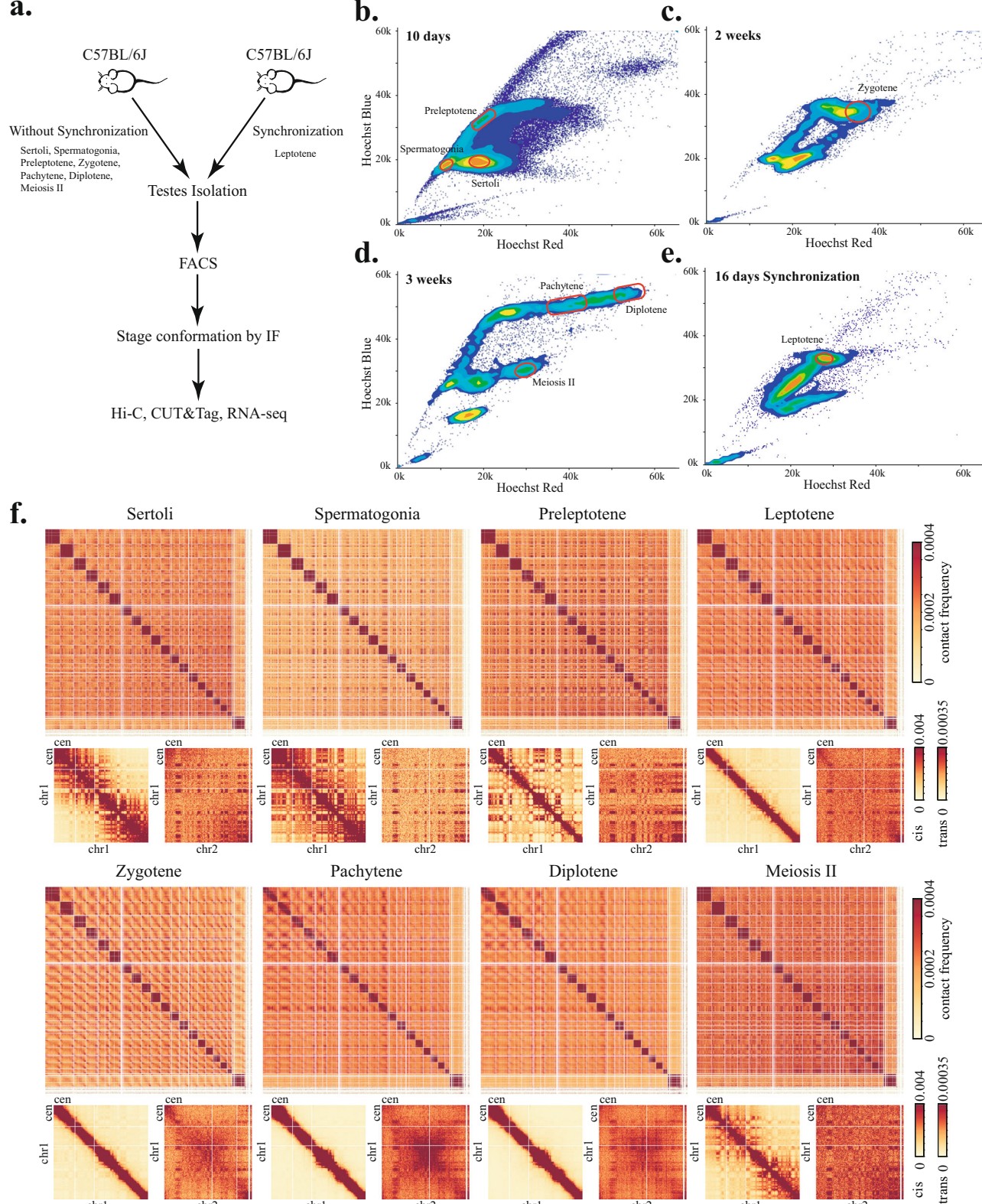

**Fig. 1 Mapping 3D genome architecture through mouse spermatogenesis by Hi-C. a** Experimental workflow for isolating somatic cells and spermatocytes of different stages. **b**–**e** Representative Hoechst profiles show the separation of different cell types by fluorescence intensity in mice of different ages: 10-day-old mice for isolating Sertoli, spermatogonia, and preleptotene cells (**b**), 2-week-old mice for zygotene cells (**c**), 3-week-old mice for pachytene, diplotene, and meiosis II cells (**d**), 16 days synchronized mice for isolation of leptotene cells (**e**). Red circles in each profile indicate gating windows used for cell isolation. **f** Genome-wide Hi-C interaction heatmaps binned at 1 Mb resolution show dynamic reorganization of 3D genome architecture during meiosis. Enlarged Chr1 and Chr1–Chr2 *trans* interaction heatmaps at 1 Mb resolution are shown below the genome-wide heatmaps. Cen indicates the centromere ends of chromosomes.

then with retinoic acid (RA)[32,40,41] (Fig. 1a, e). Evaluated by DNA content, chromatin morphology, and characteristic protein markers, the isolated cell populations were at >83% purity for leptotene- and zygotene-stage spermatocytes and >90% purity for all the other stages (Supplementary Figs. 1 and 2 and Supplementary Table 1). We further assessed the purity of the isolated cells by performing bulk RNA sequencing (RNA-seq) on two independent replicates of preleptotene-, leptotene-, and zygotene-stage spermatocytes (Supplementary Fig. 3 and Supplementary Table 2). Our purified spermatocytes exhibited characteristic transcriptional signatures that have been revealed previously by single-cell RNA-seq[42], such as Ccnd1 and Plk1 for preleptotene, Syce2 and Prdm9 for leptotene, and Sycp2 and Meiob for zygotene, further demonstrating the efficacy of our cell-isolation strategies (Supplementary Fig. 3b, c).

We performed in situ Hi-C[43,44] on two independent biological replicates for all eight isolated cell types. For each meiotic prophase I substage, we obtained a total of 200–500 million unique valid Hi-C read pairs (Supplementary Table 3). To evaluate the reproducibility of our Hi-C datasets, we calculated the Pearson correlation coefficients between the pairwise combination of datasets using the Hi-C contact probabilities binned at 500 kb resolution (Supplementary Fig. 4), as well as the insulation indices at 10 kb resolution that quantify the TAD organization (Supplementary Fig. 5). Overall, the biological replicates of the same cell type exhibit a higher degree of correlation with each other than with datasets of different cell types.

The Hi-C interaction heatmaps of different pre-meiotic and meiotic stages exhibit distinct intra- and inter-chromosome organization patterns that are consistent with previous cytological and Hi-C studies (Fig. 1f). Upon entering meiosis, the intra-chromosomal interactions become more confined near the diagonal of the heatmaps, while the checkerboard patterns off the diagonal gradually diminish, indicating the attenuation of A/B compartment organization (Fig. 1f). The inter-chromosomal interaction patterns for different meiotic substages also exhibit discernible differences, with the interactions between the central regions of different chromosomes appearing strongest at the pachytene stage (Fig. 1f)[32]. Taken together, these analyses demonstrate the good quality of our Hi-C data. The inclusion of previously unavailable preleptotene, leptotene, and diplotene stages in our cohort of Hi-C datasets provides unique opportunities for evaluating the dynamic chromosome reorganization upon meiosis entry, as well as during the progression of meiotic prophase I.

**Progressive increase of chromosome loop size throughout meiotic prophase I**. To assess how the overall chromosome organization changes during meiosis, we quantified the chromatin contact probability on autosomes as a function of genomic separation ($P(s)$) for all isolated cell populations (Fig. 2a, b and Supplementary Figs. 6–8). In Sertoli cells, the autosomal chromatin contact probability exhibits a power-law scaling with the slope of $\sim{-}1.2$ between 500 kb and 3 Mb, consistent with a chromosome folding mode of fractal globules[43] (Fig. 2a and Supplementary Figs. 6 and 7). In leptotene-stage spermatocytes, the $P(s)$ curve exhibits a shallower decay at genomic distances up to 800 kb, with a slope close to −0.6, and drops more rapidly at longer genomic distances. Such a change in $P(s)$ curve shape is in good accordance with previously reported scaling behavior for zygotene- and pachytene-stage spermatocytes[31–34] and indicates the emergence of linearly arranged chromosome loop array along chromosome axes. Interestingly, the −0.6 slope was also observed on the $P(s)$ curve of preleptotene-stage spermatocytes from 0 to

500 kb range, suggesting that chromosome loops have formed along short stretches of axial elements (Fig. 2a and Supplementary Figs. 6 and 7). Thus, the switching of chromosome folding mode has occurred prior to or at the onset of the meiosis entry.

As meiosis further progresses, the spermatocytes exhibited similar power-law decay of chromosome contacts as in the leptotene stage. However, the shallower phase of $P(s)$ curve extends to ~1.8 Mb in the zygotene stage and further to ~3 Mb in both pachytene and diplotene stages, concordant with the previous findings of progressive increase of axial chromosome compaction and chromatin loop size[32,45] (Fig. 2b and Supplementary Figs. 6 and 8).

We inferred the chromosome loop size for every stage by inspecting the relationship between the derivatives of $P(s)$ curves in log space and genomic distances. As previously demonstrated by polymer simulation, the locations of peaks on the $P(s)$ derivative plots closely match the average size of chromosome loops[46]. Using this approach, we show that the size of meiotic chromatin loops increased from ~500 kb in leptotene to ~700 kb in zygotene, eventually reaching ~1.4 Mb in pachytene and ~1.6 Mb in diplotene stages (Fig. 2c, d). Thus, the greatest increase in chromatin loop sizes occurs during the progression from zygotene to pachytene stage. Such a pattern was reproducibly observed on each autosome, as well as the X chromosome (Supplementary Figs. 9 and 10). Collectively, our analyses reveal a progressive increase of chromatin loop size throughout the entire meiotic prophase I, which may result from the loop extrusion process driven by the meiotic cohesin complex.

**TAD organization was weakened throughout the meiotic prophase I despite persistent CTCF binding**. We investigated how the switch from the fractal globule organization in interphase to linearly arranged chromosome loop arrays occurs during mammalian gametogenesis. In interphase, the chromosome fractal globules exist in the form of TADs of sub-megabase size[44,47,48]. A large fraction of TADs is demarcated by sites bound by transcription factor CTCF, which blocks cohesin-driven loop extrusion[49–51]. Thus, these TADs represent chromatin looping events between CTCF-binding sites. Previous Hi-C studies have shown that TADs were largely diminished by the zygotene stage during meiosis[32,34]. We further investigated the dynamics of TAD reorganization using our stage-resolved Hi-C datasets (Supplementary Fig. 11). Using an insulation-index approach described previously[52], we identified TAD boundaries (Supplementary Fig. 11a–f) and quantified their strength in every stage (Supplementary Fig. 11g). Consistent with the previous reports[31], our analysis indicates that the strength of TAD boundaries was already weakened in spermatogonia, which may be partially attributed to the heterogeneity in TAD organization in spermatogonia cells at different differentiation states[53]. Upon entering meiosis, the strength of TAD boundaries further decreased to about 1/2 of the somatic level in the preleptotene stage and remain attenuated throughout the rest of prophase I (Supplementary Fig. 11g–i)

The weakening of TAD boundaries throughout meiosis cannot be attributed to the loss of CTCF binding from chromosomes, which occurs during mitosis[54]. A previous chromatin immunoprecipitation–sequencing (ChIP-Seq) study on mixed pachytene/diplotene (P/D) meiocytes showed that CTCF exhibited prominent binding at >20,000 sites, a pattern similar to that in the interphase cells. We further profiled the distribution of CTCF in Sertoli cells, preleptotene, leptotene/zygotene (L/Z), and P/D spermatocytes using the CUT&Tag technique. The CTCF occupancy patterns in all four CUT&Tag datasets were similar to the P/D ChIP-Seq data (Supplementary Fig. 12a–f). Thus, although CTCF largely remained on the chromosome during

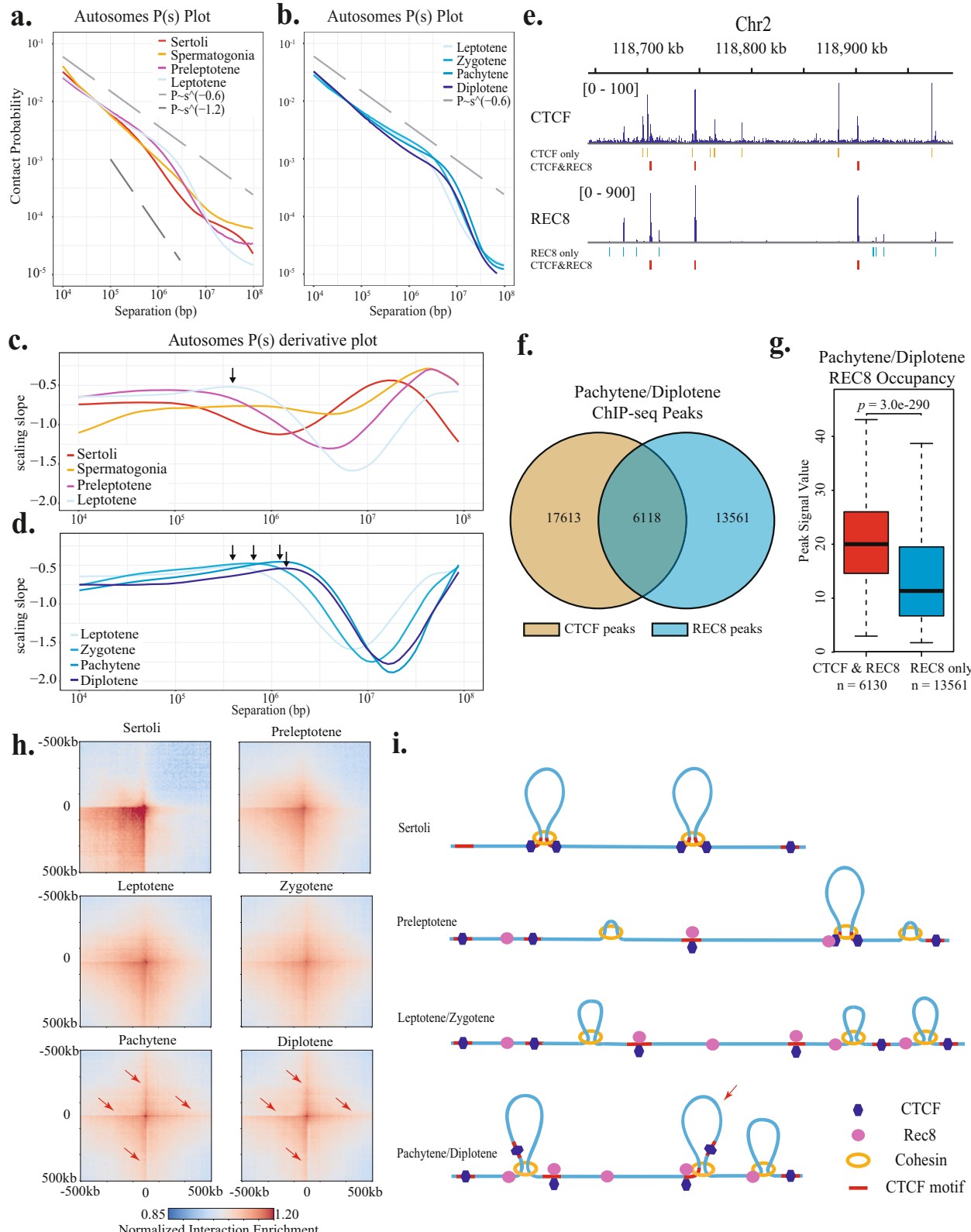

meiosis, its ability to anchor chromatin loops at specific loci and create TAD boundaries was impaired.

**Tethering chromatin loops to the meiotic chromosome axis does not create strong barriers for loop extension.** The progressive extension of meiotic chromatin loops occurs while the loops are tethered to the meiotic chromosome axis. Cohesin subunit REC8 is a major structural component of the meiotic chromosome axis. In pre-meiotic cells, REC8 already starts to assemble on chromosomes, leading to the formation of short stretches of axial elements[55]. Previously published ChIP-Seq data revealed a large number of prominent REC8 peaks in the mixed population of mouse P/D spermatocytes[34]. By comparing the

**Fig. 2 Progressive changes of chromatin loops during meiotic prophase I. a**, **b** $P(s)$ curves indicate relationships between chromatin contact probability and genomic distances for chromatin interactions on autosomes in different cell types. Dotted lines corresponding to $P(s) \sim s^{-0.6}$ and $P(s) \sim s^{-1.2}$ are shown as references. Somatic and meiotic cells exhibit different power-law scaling behavior. **c**, **d** Slopes of $P(s)$ curves at different genomic distances are used to infer the average chromatin loop size. Arrows indicate the average chromatin loop size in each meiotic prophase I substage. Chromatin loop sizes progressively increase from leptotene to diplotene stage. **e** ChIP-Seq profiles show CTCF and REC8 peaks and distribution during pachytene/diplotene stage in a representative genomic region. Bars underneath the ChIP-Seq tracks indicate locations of peaks called using MACS2. ChIP-Seq data analyzed in **e**–**g** are from Vara et al.[34]. **f** Venn diagram depicts the overlap between CTCF and REC8 peaks in pachytene/diplotene stage. **g** Boxplots indicate the distribution of ChIP-Seq peak signals for CTCF/REC8 co-occupied peaks versus REC8-only peaks in pachytene stage. The upper and lower bounds of boxes represent the third and the first quartiles of peak signal values, respectively. Centre bars represent the median peak values. The upper whisker extends from the hinge to the largest value no further than 1.5× IQR (inter-quartile range) from the hinge, and the lower whisker from the hinge to the lowest value within 1.5 × IQR of the hinge. The values beyond the whiskers are not shown in the boxplots. $n$ number of REC8 ChIP-Seq peaks ($n = 6130$ for CTCF/REC8 co-occupied peaks and $n = 13,561$ for REC8-only peaks). $p = 3.0 \times 10^{-290}$, two-sided $t$ test with the confidence level of the interval at 0.95. **h** Pileups of interactions between the 6118 CTCF/REC8 co-occupied sites and their surrounding regions within 500 kb. Bin size, 10 kb. Red arrows in pachytene and diplotene pileups indicate the cross-shaped patterns with the width of a single 10 kb bin, which represent the contacts between one CTCF/REC8 co-occupied site and one non-peak site. **i** The cartoons present a simplified view of how the chromatin loop extension is likely to occur during meiosis. In interphase cells, CTCF anchors the bases of loop domains. In preleptotene, REC8 starts to load onto chromosomes and the CTCF-anchored loops dissociate, though CTCF remain largely bound to the chromosomes. In leptotene/zygotene and pachytene/diplotene, cohesin-driven loop extrusion leads to the progressive increase of loop sizes. The REC8-binding sites do not constitute strong barriers for loop extension and can be skipped. Therefore, unlike in interphase, the meiotic chromatin loops are heterogeneously distributed and not anchored at fixed positions. Note that the binding of CTCF and REC8 depicted in these illustrations reflects the population-averaged protein occupancy, not individual binding events in single cells.

stage-matched REC8 and CTCF ChIP-Seq profiles, we found that about one-third of REC8 peaks colocalize with CTCF peaks (Fig. 2e, f). Notably, the 6130 REC8 peaks that overlap with CTCF peaks in P/D (CTCF/REC8 co-occupied peaks) exhibit nearly twofold higher REC8 occupancy than the 13561 REC8 peaks that do not coincide with CTCF peaks (Fig. 2g, $p = 3.0 \times 10^{-290}$, two-sided $t$ test, Supplementary Fig. 12g). Although the binding of REC8 in mammals is likely to be stochastic among cells, these CTCF/REC8 co-occupied peaks may represent locations where the chromatin is associated with the axis at a higher frequency. Moreover, CTCF occupancy at these sites persists through the meiosis (Supplementary Fig. 12b–f). Thus, these CTCF/REC8 co-occupied peaks provide a representative set of locations for evaluating how tethering to chromosome axis influences the behavior of meiotic chromatin loops.

We quantified the insulation indices of 10 kb genomic intervals coincident with different categories of peaks before and after meiosis entry. The insulation index reflects the aggregated interactions occurring across each genomic interval. Thus, lower insulation scores associated with genomic loci indicate stronger insulation ability. In Sertoli cells, the genomic regions coincident with either the CTCF/REC8 co-occupied peaks or the CTCF-only peaks exhibited significantly lower insulation scores than the regions not associated with CTCF or REC8 peaks (Supplementary Fig. 13a). Upon the meiosis entry, the insulation scores associated with all the peaks elevated compared to the non-peak regions (Supplementary Fig. 13b), with the greatest elevation of insulation scores occurring at the CTCF/REC8 co-occupied sites (Supplementary Fig. 13c), indicating the greatest loss of insulation abilities at these locations. Such a pattern persisted through the rest of meiosis (Supplementary Fig. 13d–g).

We further analyzed the positioning and distribution of chromatin loops using composite pileups of interactions among the 6118 CTCF/REC8 co-occupied sites. In interphase cells, interactions among these sites are highly enriched, as indicated by the strong signals at the center of pileup heatmaps (Fig. 2h, i). These interactions substantially decreased in preleptotene cells, indicating the disruption of CTCF-anchored loops (Fig. 2h, i). As meiosis progresses, chromatin loops extend to sizes >1 Mb, which exceeds both the average size of TADs in interphase and the ~120 kb median separation between the neighboring CTCF/REC8 co-occupied sites. If these axis-tethering sites possess the ability to halt the extension of loops, the population-averaged interaction

frequencies among them would gradually increase even though the positions of loops vary from cell to cell. However, such an increase of contact was not observed in late meiotic prophase, suggesting that these sites do not confer strong constraints on chromatin loops and could be bypassed during the loop extension (Fig. 2h, i). Interestingly, in the pachytene and diplotene stages, the signals along the narrow strips extending from the center of pileup heatmaps became more evident compared to the early meiotic prophase (arrows, Fig. 2h). Such a "cross-shaped" pattern indicates a slight enrichment of chromatin loops between one CTCF/REC8 site and one non-CTCF/REC8 region (arrow, Fig. 2i). Thus, it remains possible that the CTCF/REC8 sites could function as loop-extrusion barriers but to a lesser extent than the CTCF binding sites in interphase. Nonetheless, our data suggest that the high occupancy of CTCF and REC8 at these sites do not create a strong barrier for meiotic loop extension. The inability to immobilize chromatin loops at specified locations may contribute to heterogeneity in meiotic chromatin loop sizes during the progression of meiosis.

**Dynamic changes of chromatin organization at the DSB hotspots during meiotic prophase I.** During the early meiotic prophase, DSBs form within the axis-attached chromatin loops and promoted the co-alignment of homologs. Our Hi-C data at higher temporal resolution enable the characterization of the dynamic chromatin organization around the DSBs throughout meiosis. We derived the positions of DSB hotspots using the previously published DMC1 single-stranded DNA ChIP data[56], classified these sites into either CO-favored or CO-disfavored DSB hotspots (CO-DSBs and NCO-DSBs) based on their distances to the previously identified CO hotspots[30], and examined the chromatin organization patterns surrounding each class of sites during the meiosis progression using Hi-C interaction pileups (Fig. 3a–c).

The CO-DSBs and the NCO-DSBs exhibit distinct Hi-C interaction patterns compared to the control sites not associated with DSBs in both somatic and meiotic cells. In Sertoli cells, the 2 M genomic neighborhoods surrounding both the CO-DSBs and NCO-DSBs exhibit elevated chromatin interactions than the control regions, with the CO-DSB regions exhibiting the highest interactions (Fig. 3a, b). Moreover, a weak, cross-shaped pattern extending from the CO-DSBs was observed on the pileup

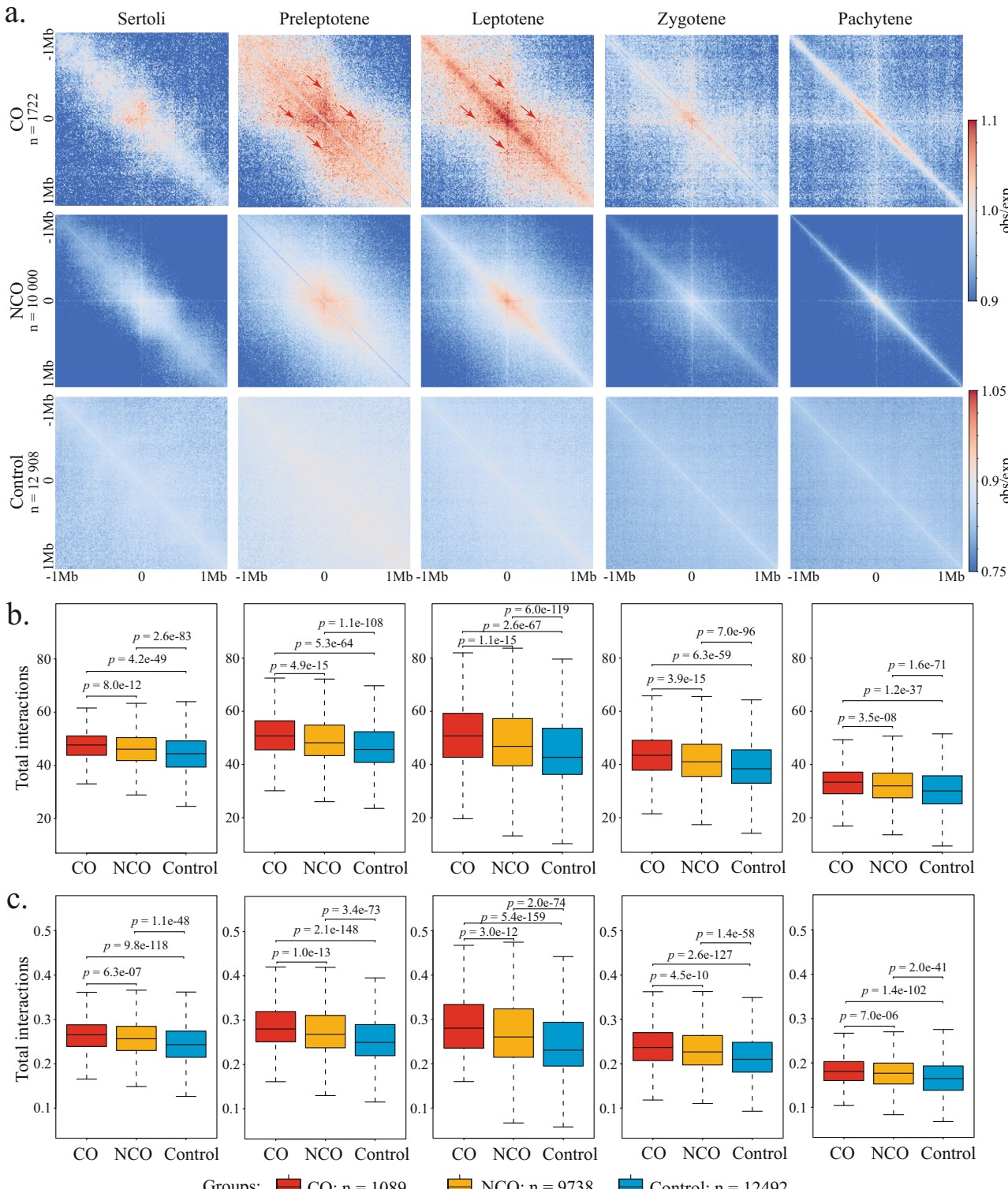

**Fig. 3 DSB hotspots exhibit distinct chromatin organization during meiosis prophase I. a** Pileup heatmaps of 2 Mb genomic regions centered at CO-DSBs (top row), NCO-DSBs (middle row), and non-DSB sites (bottom row) in different cell types. Red arrows indicate the cross-shaped pattern extending from the CO-DSBs in preleptotene and leptotene. The observed Hi-C interaction frequencies are normalized using expected interaction frequencies at each genomic distance (Obs/Exp). Bin size, 10 kb. **b** Box plots compare the total balanced Hi-C interactions in 2 Mb genomic regions centered at CO-DSBs (red), NCO-DSBs (orange), and non-DSB sites (blue) in different stages. **c** Box plots compare the total balanced Hi-C interactions between the single 10 kb genomic bins containing the CO-DSBs (red), NCO-DSBs (orange), and non-DSB sites (blue) and the 1 Mb flanking regions on each side in different stages (i.e., the total signals along the center line in the Hi-C heatmap for each 2 Mb genomic region in **b**). For boxplots in **b**, **c**, center line, median; box limits, upper and lower quartiles; whiskers, 1.5× interquartile range. The values beyond the whiskers are not shown in the boxplots. p values indicated in **b**, **c** are calculated from one-tailed Mann–Whitney U test. n, total number of the 10 kb bins containing CO-DSBs (n = 1089), NCO-DSBs (n = 9738), and non-DSB (n = 12492) sites.

heatmap in Sertoli cells (Fig. 3a), suggesting that the CO-DSBs preferentially interact with their flanking regions compared to the NCO-DSBs and the control sites. Further quantification of the interactions between the 10 kb genomic bins containing different classes of sites and their flanking bins corroborates such a notion (Fig. 3c). The elevated interactions within the genomic neighborhoods of CO-DSBs and between the CO-DSBs and the flanking regions persist through meiosis (Fig. 3b, c).

Notably, the DSB regions, particularly the ones associated with CO-DSBs, exhibit pronounced changes in chromatin interaction patterns during early meiotic prophase I, when homolog pairing and co-alignment occurs. During preleptotene and leptotene stages, the CO-DSBs, but not the NCO-DSBs, exhibit a transient, TAD-boundary-like pattern (Fig. 3a). Concomitantly, the cross-shaped patterns became more prominent and extended over longer distances at the CO-DSBs (arrows, Fig. 3a). These unique intra-chromosomal interaction patterns may reflect the dynamic behaviors of DSB sites during homology search. Together, these findings suggest that the higher-order chromatin organization surrounding the DSB sites correlate with their future fates of whether to be resolved as COs or non-COs.

**Transcription-coupled variations in chromatin loop sizes during early meiotic prophase.** One distinct feature of meiotic chromosomes compared to mitotic chromosomes is their persistent transcriptional activity. We examined whether the meiotic loop array organization could be influenced by differential levels of transcription along chromosomes. In interphase cells of mammals, the transcriptionally active and inactive genomic regions preferentially associate with regions of similar transcriptional status, forming spatially segregated A and B compartments[43]. Several recent Hi-C studies have revealed that the A/B compartment organization is maintained throughout meiotic prophase I[31–34]. Our Hi-C data at higher temporal resolution corroborate such a notion (Fig. 4 and Supplementary Figs. 14, 15). Using our bulk RNA-seq datasets for preleptotene-, leptotene-, and zygotene-stage spermatocytes, we showed that the A/B compartment identified by principal component analysis exhibited a good correlation with the transcriptional activities (Supplementary Fig. 14). A majority of A and B compartment regions retain their respective compartment identity throughout the entire meiotic prophase I (Supplementary Fig. 15a). Interestingly, after the meiosis entry, the fraction of genome belonging to active A compartment first decreases in leptotene and zygotene stages and subsequently increases in pachytene and diplotene stages (Supplementary Fig. 15a, b), in line with the previously reported elevation in total transcription level at the pachytene stage[57]. Upon meiosis entry, the extent of genome compartmentalization, which is indicated by the checkerboard pattern on Hi-C heatmaps, exhibited a marked decrease in leptotene and zygotene and further diminished in pachytene and diplotene stages (Fig. 4a). Such an attenuation in compartments may result from the compaction of chromosome loop arrays and the global loss of long-range chromatin interactions. Whereas the long-range association between genomic regions belonging to the same compartment decreased, the interactions between regions belonging to different compartments remained at a low level (Supplementary Fig. 15b, c). These findings suggest that the transcriptionally active and inactive regions form alternating, large-scale chromatin domains along the meiotic chromosome axes, consistent with the previous cytological observations[4].

The transcriptionally active and inactive domains exhibit distinct chromatin folding patterns in early meiotic prophase. In preleptotene-, leptotene-, and zygotene-stage spermatocytes, transcriptionally active A compartment regions exhibited markedly higher chromatin interaction frequencies at shorter genomic distances than inactive B compartment regions, resulting in darker segments along diagonal on Hi-C heatmaps (Fig. 4a, b). Such a transcriptional status-coupled difference in chromatin interactions can be more easily appreciated upon normalization using expected interaction frequency at every genomic distance (Fig. 4c). Quantification of observed chromatin interaction frequencies reveals that, during leptotene and zygotene stages, loci within A compartment regions interact at least 50% more frequently at 100–500 kb genomic separation than loci within B compartment regions (Fig. 4d). In contrast, the B compartment regions exhibited higher interaction frequencies at longer genomic distances. We note that such a disparity in chromatin interactions in the early meiotic prophase is unlikely caused by the regional variations in inter-homolog interactions, as the differences between A and B compartment regions are more prominent in the leptotene stage, when synapsis has not initiated, than in zygotene stage. During later stages of the meiotic prophase, the transcription-coupled differences in chromatin interactions become less pronounced (Fig. 4d).

The enrichment of short-range interaction for A compartment domains suggests these regions consist of shorter chromatin loops compared to the transcriptionally inactive regions. We examined the $P(s)$ relationships for genomic regions >2 Mb that belong to either A or B compartment and estimated the average chromatin loop sizes in these regions using the slope of $\log(P(s))$. The average loop sizes in A and B compartment regions are 560 kb versus 730 kb during the leptotene stage, respectively, and increase to 800 kb versus 1.05 Mb in the zygotene stage (Fig. 4e). Such a difference in loop size could be attributed to the higher density of loops pre-existing prior to meiosis entry, as well as the slower rate of chromatin loop extension, in actively transcribed genomic regions. In later prophase, as loops continue to extend, the chromatin loops for A and B compartment regions eventually reached similar sizes (Fig. 4e). Taken together, our data unveil that the domanial organization of transcription activities along the chromosome axis correlates with variations in chromatin loop properties during the early stages of meiotic prophase I (Fig. 4f), when homolog pairing and alignment take place.

**Variations in chromatin loop size along chromosome axis correlate with the extent of homolog alignment and the probability of CO formation.** The disparity in chromatin loop sizes for A/B compartment domains during early meiotic prophase could profoundly influence contacts between homologs. The shorter loop size for A compartment regions may result in a higher number of loops for the same amount of DNA content and a higher fraction of DNA located in the spatial vicinity of chromosome axes. Such a conformation could conceivably increase the probability for DNA to access recombination machinery, thereby promoting meiotic recombination and facilitating homolog recognition.

To test the above hypothesis, we re-analyzed previously published Hi-C data of zygotene-stage spermatocytes from B6/CAST hybrid mice[32] to assess the relationships between compartment identity, chromatin loop size, and inter-homolog alignment (Fig. 5). To evaluate the extent of homolog alignment during the zygotene stage, we devised "alignment scores" by calculating the total interaction frequency for every genomic interval with the regions on its homologous chromosome within 100 kb. We found that genomic intervals belonging to the A compartment exhibit significantly higher alignment scores than the B compartment, indicating a greater extent of alignment for transcriptionally active regions (Fig. 5a, b). We further divided the A compartment regions >1 Mb in zygotene into three groups (A1, A2, and A3) based on the

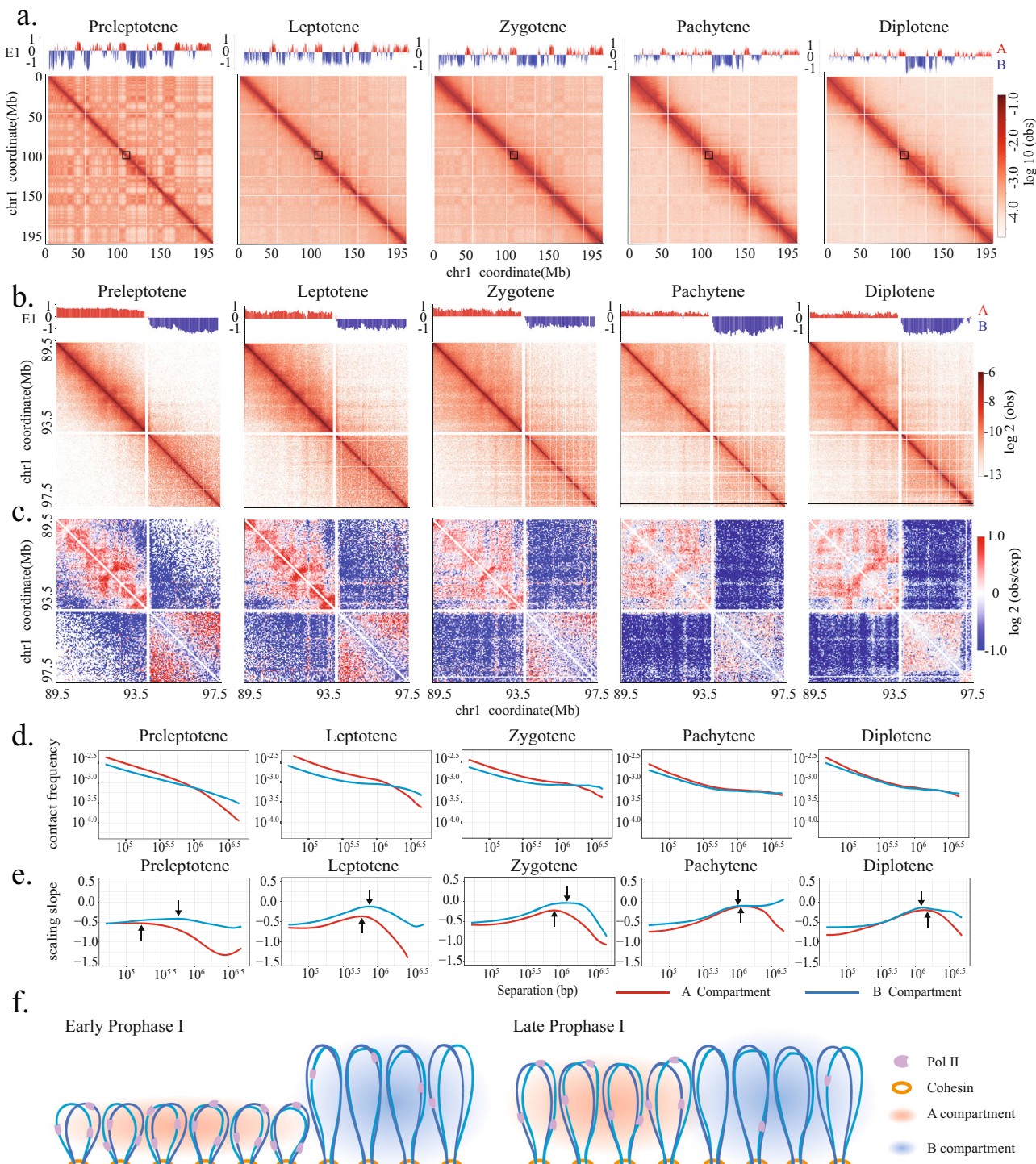

**Fig. 4 Transcription-coupled variations in chromatin loop organization during early meiotic prophase. a** Chr1 heatmaps binned at 50 kb resolution show attenuation of compartmentalization during meiotic prophase I. Plots of eigenvector 1 values on top of heatmaps indicate positions of A (red) and B (blue) compartments. The compartment identity is largely preserved throughout meiotic prophase I. Arrowheads indicate segments of thickened signals along diagonals, which coincide with A compartment. Close-up views of boxed regions are shown in **b**, **c**. **b**, **c** Heatmaps of observed interaction frequency (**b**) and observed/expected interaction ratio (**c**) at 20 kb resolution for a representative Chr1 region. In preleptotene, leptotene, and zygotene stages, the A and B compartment regions enrich for interactions at shorter and higher genomic separation, respectively. **d** $P(s)$ plots show relationships between chromatin contact frequency and genomic separation for A (red) and B (blue) compartment regions during different meiotic stages. Only A and B compartment regions >2 Mb were included in this analysis. **e** Plots of $P(s)$ slopes show chromatin loop sizes for A and B compartment regions in each meiotic prophase I substage. Arrows indicate the peak positions in $P(s)$ slope curves, which correspond to the average loop sizes. The A compartment regions consist of smaller loops than the B compartment regions in preleptotene, leptotene, and zygotene stages. **f** Schematic illustrations show that the transcriptionally active and inactive regions exhibit different chromatin loop sizes during early meiotic prophase I. The differences become less pronounced in late meiotic prophase I.

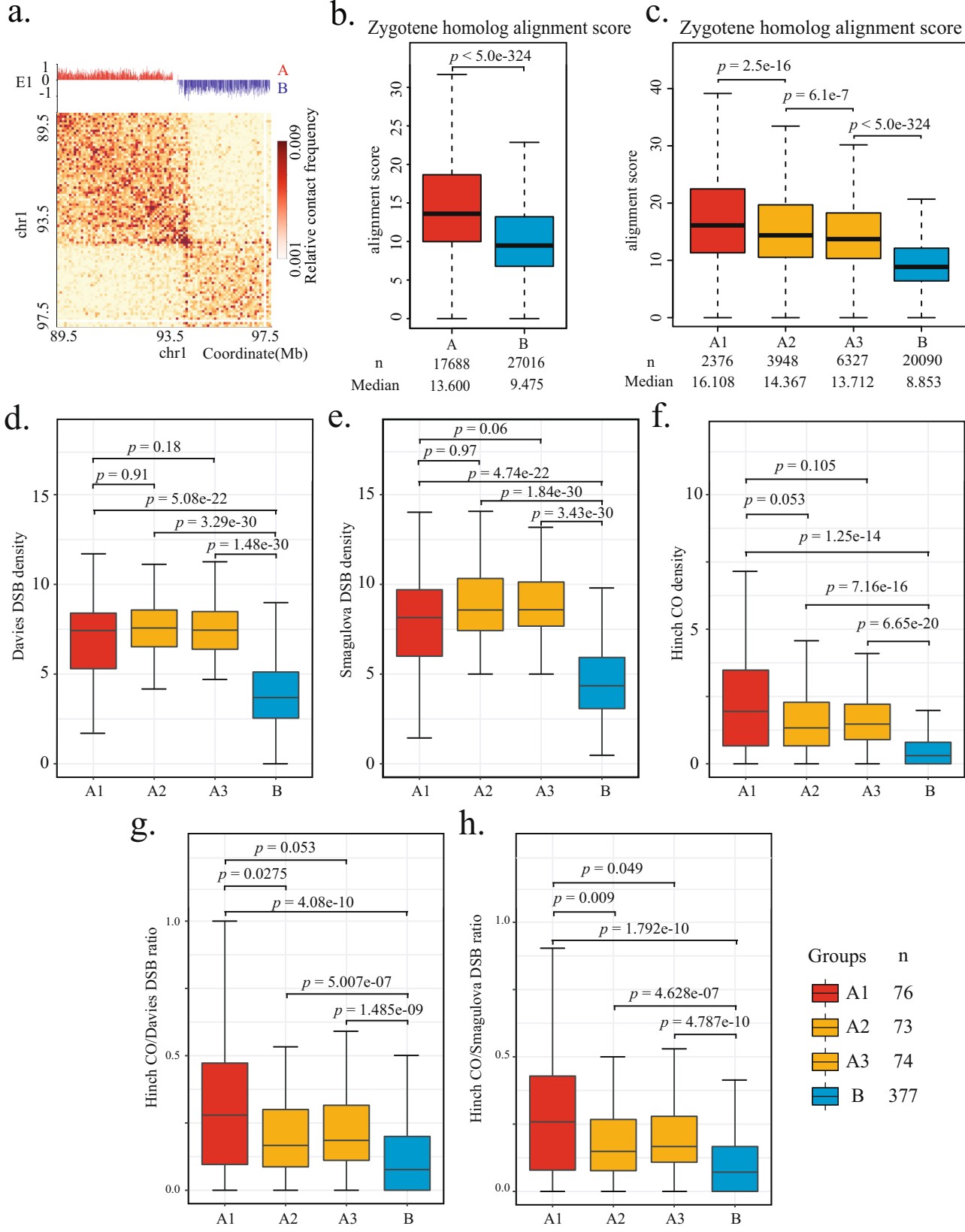

chromatin interaction frequency at 500 kb genomic separation, which serves as an indicator for the enrichment of shorter chromatin loops. Importantly, the genomic intervals within the A1 group regions, which show the greatest enrichment of short chromatin loops, also exhibited the highest alignment scores (Fig. 5c). These

analyses implicate the transcription-coupled variations in chromatin loop size in the modulation of homolog alignment during early meiotic prophase.

We also found a link between the chromatin loop sizes and the formation of COs. We calculated the density of DSBs per mega

**Fig. 5 Variations in intra-chromosomal loop organization correlate with homolog alignment and crossover distribution. a** Z-stage inter-homolog interaction heatmap at 50 kb resolution for the same region shown in Fig. 4b, c. The inter-homolog interaction data are from Patel et al.[32]. **b** Boxplots quantify distributions of inter-homolog alignment scores for 50 kb genomic intervals belonging to A (red) or B (blue) compartment. *n*, total number of 50 kb intervals for A (*n* = 17,688) and B (*n* = 27,016). *p* value is calculated from two-tailed Mann–Whitney *U* test. **c** The A compartment regions >1 Mb are ranked based on the averaged interaction frequencies at 500 kb and divided into three groups (A1, A2, A3). Boxplots depict distributions of inter-homolog alignment scores for 50 kb genomic intervals belonging to A1 (red), A2 (orange), A3 (orange), or B (blue) compartment. *n*, total number of 50 kb intervals for each category (A1, *n* = 2376; A2, *n* = 3948; A3, *n* = 6327; B, n = 20,090). *p* values are calculated from two-tailed Mann–Whitney *U* test. **d–h** Boxplots quantify distributions of DSBs, COs, and the CO/DSB ratios in A1, A2, A3, and B compartment regions >1 Mb. The positions of DSBs and COs are from Davies et al.[56], Smagulova et al.[58], and Hinch et al.[30]. **d, e** A1 regions do not exhibit significantly higher DSB density than A2 and A3 regions. **f** A1 regions do not exhibit significantly higher crossover densities than A2 and A3 regions. **g, h** A1 regions exhibit significantly higher crossover/DSB ratio than the A2 regions. *n*, total number of A1 (*n* = 76), A2 (*n* = 73), A3 (*n* = 74), and B (*n* = 377) regions. *p* values in **d–h** are calculated from one-tailed Mann–Whitney *U* test. For boxplots in **b–h**, center line, median; box limits, upper and lower quartiles; whiskers, 1.5× interquartile range. The values beyond the whiskers are not shown in the boxplots.

base pairs for each A and B compartment region >1 Mb using the DMC1 single-stranded DNA ChIP data from two independent studies[56,58] and also calculated the CO density using the data generated by single-sperm sequencing. As shown previously, the A1, A2, and A3 regions all exhibit significantly higher DSB densities and CO densities compared to the B compartment regions (Fig. 5d–f), consistent with the notion that the DSB events are mainly influenced by local chromatin properties such as GC content and histone modifications[59]. Notably, all A compartment regions also exhibit significantly higher CO/DSB ratios than the B compartment regions (Fig. 5g, h), suggesting that the DSBs within the A compartment regions are more likely to mature into COs. Interestingly, we found that, whereas the A1 regions enriching in short chromatin loops do not exhibit higher DSB or CO densities than the other A compartment regions (Fig. 4d–f), they do exhibit higher CO/DSB ratios than the A2 regions (Fig. 5g, h). Together, these results suggest that the enrichment of short chromatin loop sizes is associated with a higher conversion rate of DSBs to COs. Therefore, the chromatin loop sizes along the meiotic chromosome axis may contribute to the modulation of meiotic recombination landscape.

**Alignment of chromosome ends over a substantial range during early meiotic prophase.** In addition to the changes in intra-chromosomal loop organization, meiotic chromosomes also undergo dramatic repositioning relative to each other during early prophase I to facilitate homolog juxtaposition and alignment[21]. We assessed the temporal pattern of chromosome end association during meiotic prophase I. The 5 Mb sub-telomeric regions at both centromere-proximal and centromere-distal ends exhibit prominent, polarized associations during the leptotene stage. The association among chromosome ends is most prominent during the zygotene stage and drops to the same level as in interphase cells during pachytene and diplotene stages (Fig. 6a–e and Supplementary Fig. 16). Such a pattern is highly concordant with the timing of the nuclear peripheral targeting and clustering of telomeres from cytological studies[60].

Remarkably, we observed that large portions of chromosomes near telomeres on different chromosomes tend to align with each other during early meiotic prophase. To quantitatively characterize such a pattern, we averaged the heatmaps for all 171 pairwise combinations of autosomes after normalization for different chromosome lengths. The averaged heatmaps reveal that these "alignment tracts" among different autosomes persist over a substantial range of ~20% of chromosome length from either chromosome end in both leptotene and zygotene stages (Fig. 6e, f) but largely diminished in late meiotic prophase (Fig. 6e and Supplementary Fig. 17).

We note that, although the degree of inter-chromosomal alignment was the highest during the zygotene stage (Fig. 6e and

Supplementary Fig. 16d), the shapes of alignment tracts were largely identical for leptotene and zygotene (Fig. 6e, f). In most of our isolated zygotene spermatocytes, meiotic chromosomes appeared dispersed or formed several small clusters, each of which consists of multiple chromosomes (Supplementary Fig. 18a, b, d). Only <5% of zygotene-stage spermatocytes exhibited obvious bouquet arrangement, in which a large number of chromosome ends coalesce and exhibit polarized localization (Supplementary Fig. 18c). Thus, the prominent inter-chromosomal alignment near chromosome ends observed in our Hi-C data is not merely a manifestation of the bouquet conformation. Instead, it indicates that, whenever different chromosomes are brought into proximity by telomeres during early prophase I, a substantial fraction of chromosomes adjacent to telomeres also tend to align with each other. Such a phenomenon implies that the genomic regions near chromosome ends may adopt a special configuration that favors their alignment.

Quantification of zygotene-stage inter-homolog interactions revealed a pattern largely concordant with the interactions among non-homologous chromosomes (Fig. 6g). Such a telomere proximity-dependent pattern is independent of compartment identity. Importantly, the windows of elevated inter-chromosomal alignment also coincide with regions exhibiting high CO frequency[30,61], suggesting that the spatial arrangements of chromosome ends may play a part in the regulation of meiotic recombination.

To assess whether the modulation of chromatin loops is involved in the prominent alignment of chromosome ends, we compared average loop sizes for both A and B compartment regions located at chromosome ends versus the middle of chromosomes (Supplementary Fig. 19). For transcriptionally active A compartment regions, no differences in loop sizes were observed between chromosome ends and central regions (Supplementary Fig. 19a, b). On the other hand, transcriptionally inactive B compartment regions located at the centromere distal but not the centromere-proximal ends of chromosomes exhibited a smaller loop size than the rest of the chromosome (Supplementary Fig. 19d, e). We note that only a small fraction of genomic regions near the centromere-distal ends of chromosomes belong to the B compartment. Thus, the difference in chromatin loop size is unlikely the main cause for prominent alignment among chromosome ends.

**Extensive alignment of chromosome ends depends on the association of telomeres with the LINC complex.** We next sought to investigate the underlying mechanisms for the extensive chromosome end alignment during early meiotic prophase. In one possible scenario, the association of telomeres at the nuclear periphery would automatically confer the alignment of the sub-

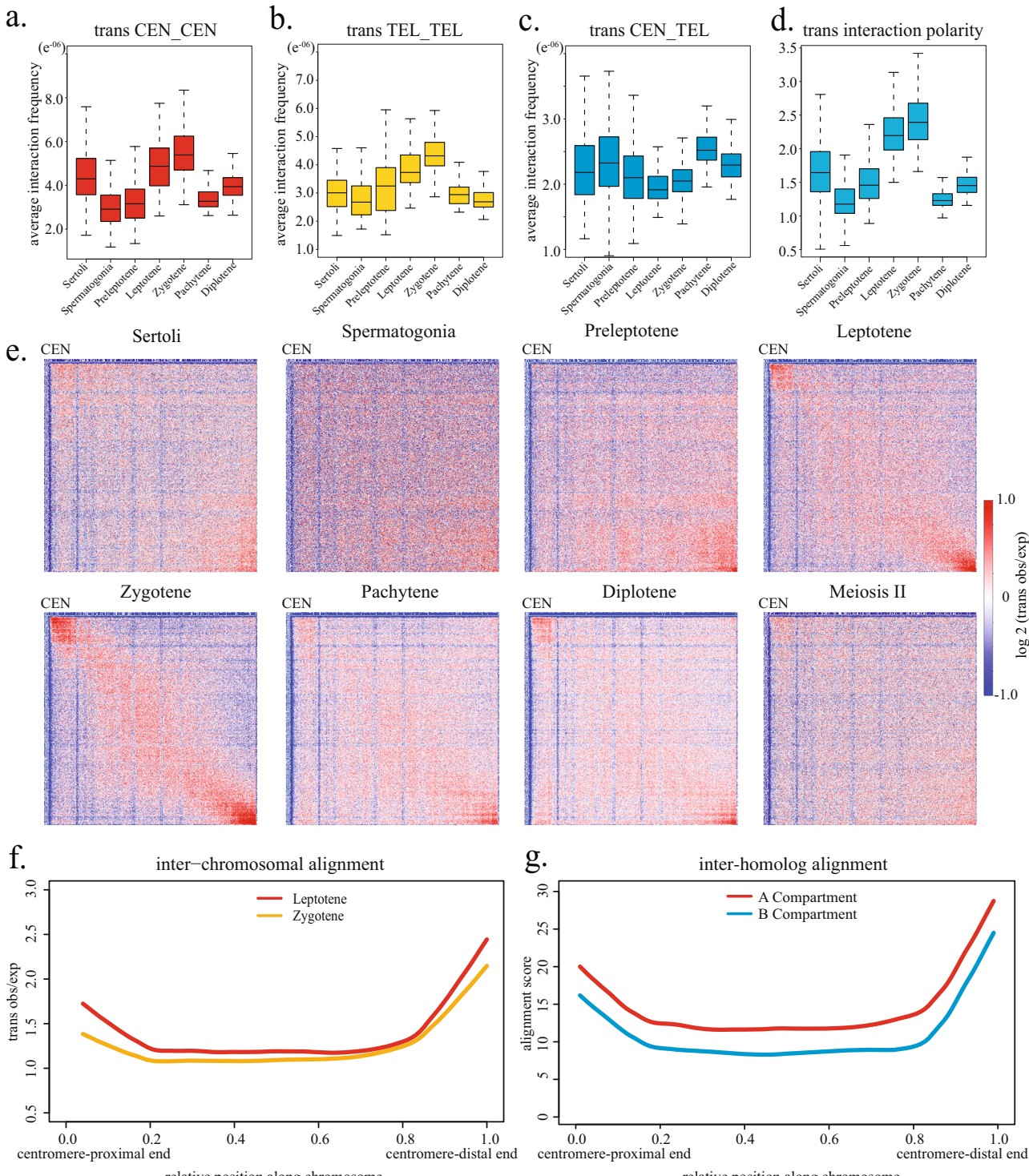

telomeric regions of ~20% chromosome lengths. Alternatively, other extrinsic factors or mechanisms may remodel the conformation of chromosome ends to promote their alignment over a substantial range. One such mechanism could involve the LINC complex, which transmits force from the cytoskeleton to telomeres.

To distinguish between these two possibilities, we took advantage of a mouse model defective in the telomere–LINC association. We recently discovered that the association between telomeres and the LINC complex is mediated by the protein–protein interaction between SUN1 and telomere-associated protein Speedy A

(SPDYA). Disrupting the SUN1–SPDYA interaction interface in mice by a single point mutation (SUN1 W151R) abolished telomere–LINC association, resulting in severe homolog pairing defects, complete meiotic arrest, and sterility[62]. However, the nuclear peripheral anchoring of telomeres was only partially abrogated in Sun1^W151R/W151R mutant mice, with about 50% of telomere remaining anchored to the INM at the zygotene stage (Supplementary Fig. 20). Consistent with the cytological observations, we have previously shown by Hi-C that the Sun1^W151R/W151R mutant spermatocytes exhibited a significant decrease, but not a complete loss, of the interactions among the telomeres[62]. Such a

**Fig. 6 Extensive alignment of chromosome ends during early meiotic prophase. a–c** Boxplots quantify interactions among the 5 Mb sub-telomeric regions on different chromosomes. Each box shows the distribution of average interaction frequency per genomic 50-kb bin in sub-telomeric regions for 171 pairwise combinations of autosomes. Leptotene and zygotene stages exhibit the highest interactions between sub-telomeric regions at the centromere-proximal ends of different chromosomes (CEN-CEN, **a**) and between the centromere-distal ends of different chromosomes (TEL-TEL, **b**), but the lowest interactions between sub-telomeric regions at different chromosome ends (CEN-TEL, **c**). Box limits, upper and lower quartiles. Centre bars, median. Whiskers, 1.5× interquartile range. **d** Boxplots quantify the polarity of interactions among the sub-telomeric regions by calculating (CEN-CEN + TEL-TEL)/ (2 × CEN-TEL) for each of 171 pairwise combinations of autosomes. Leptotene and zygotene stages exhibit the highest polarity. For all boxplots in **a–d**, $n = 171$ pairwise combinations of autosomes. **e** Averaged *trans* observed/expected heatmaps of all 171 pairwise combinations of autosomes. Interpolation was performed to normalize for different chromosome lengths. Matrices were scaled to 500 bins × 500 bins. **f** Plots quantifying the signals along the diagonals of the heatmaps **e** indicate the extent of inter-chromosomal alignment along chromosomes. Regions of ~20% chromosome length from either chromosome end exhibit prominent inter-chromosomal alignment in leptotene and zygotene stages. **g** Plots quantify zygotene-stage inter-homolog alignment scores versus chromosome position for A and B compartment regions. Inter-homolog alignment patterns are similar to the inter-chromosomal alignment patterns in **f**.

model provides unique opportunities to uncouple the effects of LINC-mediated force transmission on chromosome organization from the effects of telomere peripheral targeting.

We carefully compared the Hi-C map from Sun1$^{W151R/W151R}$ mutant zygotene spermatocytes (Fig. 7a, b) with the maps of precisely staged wild-type spermatocytes generated in this study using the same experimental procedures. The Sun1 mutant zygotene spermatocytes exhibit the highest correlation with the wild-type zygotene spermatocytes in both Hi-C contact probabilities and insulation indices (Supplementary Figs. 4 and 5). The Sun1 mutant zygotene spermatocytes exhibited minimal changes in $P(s)$ shape compared to wild-type zygotene spermatocytes (Fig. 7c). Furthermore, the average chromatin loop sizes across different chromosome positions exhibited no changes in Sun1$^{W151R/W151R}$ mutant (Supplementary Fig. 19c, f). Therefore, the tethering of telomeres to the LINC complex does not significantly affect intra-chromosome organization.

If the extensive alignment of chromosome ends over a long range does not depend on the LINC complex, we would expect the mutant inter-chromosome alignment profile to exhibit a similar shape to the wild-type profile. However, the pattern of inter-chromosome alignment drastically changed. We found that the telomeres still exhibit substantial association in Sun1 mutant spermatocytes. Notably, the association among the very tips of chromosomes even increased in Sun1 mutant compared to the wild-type spermatocytes. On the other hand, the tracts of alignment dropped abruptly after extending for a range of ~5% of chromosome length, greatly shortened compared to the ~20% chromosome length in wild-type leptotene and zygotene spermatocytes (Fig. 7d–f). Thus, in the absence of the association with the LINC complex, different chromosomes could still be brought to proximity due to the nuclear peripheral targeting of telomeres, but the contacts at the chromosome ends persist over a shorter fraction of chromosome length (Fig. 7g).

Taken together, our observations suggest that the tethering of telomeres to the LINC complex, and the subsequent force transmission, plays an essential role in promoting the "tight" alignment of chromosome ends. Such a function by the LINC complex to regulate spatial chromosome organization may promote homolog synapsis and CO formation.

## Discussion

During meiosis, chromosomes reorganize into sequential loop arrays along the chromosome axis. Although several recent Hi-C studies explored the functional relationships between meiotic chromosome organization and transcriptional regulation[31–34], how the highly specialized meiotic chromosome structure impacts homolog interactions and meiotic recombination is less clear. In this study, we systematically characterized meiotic

chromosome structure throughout all substages of meiotic prophase I. Our analyses reveal chromosome organization features that correlate with the pattern of homolog alignment during early meiotic prophase, suggesting a functional link between the chromosome structure properties and the regulation of the meiotic program.

Our Hi-C datasets at higher temporal resolution revealed a progressive increase of chromosome loop size from ~500 kb to 1.6 Mb during meiotic prophase I, consistent with the previous findings in zygotene and pachytene spermatocytes[32]. While the exact molecular mechanisms underlying the extension of meiotic chromatin loops have not been demonstrated, this process may be explained by a cohesin-driven loop extrusion model, which serves as a pivotal mechanism in both interphase and mitotic chromosome organization. In interphase cells, the loop extrusion by cohesin was halted by CTCF, resulting in a high frequency of interactions between CTCF-binding sites and the formation of TADs. However, TADs are weakened and the looping interactions between reproducible locations are notably absent during meiosis in mammals. Here we show that such an effect cannot be attributed to the dissociation of CTCF from meiotic chromosomes. We further show that a set of CTCF-binding sites exhibit high levels of REC8 binding during meiosis, implying that these locations are associated with the chromosome axis at a high frequency. However, these "tethering sites" do not exhibit increased interactions as meiosis progresses, suggesting that neither the binding of CTCF nor REC8 can create efficient barriers to halt the extension of loops. Such a lack of structural constraints may contribute to the flexible regulation of meiotic chromatin loop sizes.

Chromatin organization has long been linked to CO modulation in mammals. In both mouse and human, females exhibit higher CO frequency than males while also exhibiting longer SC and shorter chromatin loops[17,63,64]. Another type of sexual dimorphism in CO frequency has been observed in the short pseudo-autosomal region (PAR) that mediates the pairing of sex chromosomes. In male mice, the PAR exhibits elevated CO frequency than in females, coincident with an extended conformation consisting of significantly shorter loops[65,66]. Moreover, a recent study revealed that the frequencies of COs covary across different chromosomes in individual nuclei[67]. Importantly, the CO covariation can be explained by the per-nucleus covariation in chromosome axis lengths, strongly suggesting the regulation of axis length and/or chromatin loop size may act as a general mechanism for the global modulation of COs[68]. It is thus conceivable that the regional variations in chromatin loop sizes and axis lengths may influence the distribution of CO events across the genome, resulting in the enrichment of recombination and COs in transcriptionally active, gene-rich regions. Although such a model is attractive, numerous cytological studies on pachytene

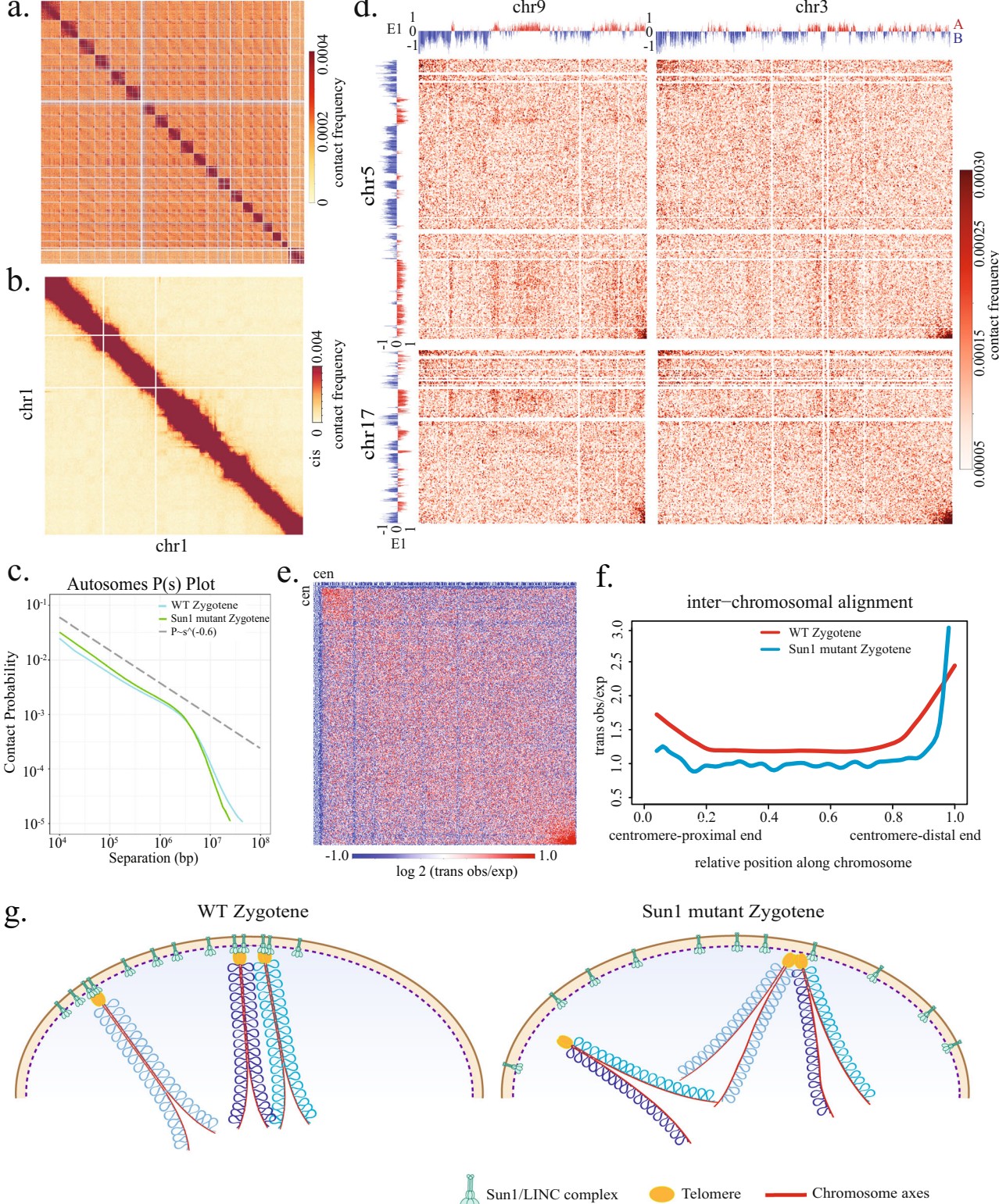

**Fig. 7 Telomere–LINC complex association is required for chromosome end alignment over a substantial range. a** Genome-wide Hi-C interaction heatmaps binned at 1 Mb resolution for Sun1[W151R/W151R] mutant zygotene spermatocytes. **b** Chr1 interaction heatmaps binned at 1 Mb resolution for Sun1[W151R/W151R] mutant zygotene spermatocytes. **c** P(s) curve of Sun1[W151R/W151R] mutant zygotene spermatocytes exhibits a similar shape to wild-type zygotene spermatocytes. **d** Interaction heatmaps at 500 kb resolution showing *trans* observed interaction frequencies among chromosomes 3, 5, 9, and 17. Different chromosomes only exhibit association at the very tips of chromosomes. **e** Averaged *trans* observed/expected heatmaps of 171 pairwise combinations of autosomes in Sun1[W151R/W151R] mutant zygotene spermatocytes. **f** Plots compare signals along the diagonals of the averaged *trans* heatmaps for wild-type and Sun1[W151R/W151R] mutant zygotene spermatocytes. The inter-chromosomal alignment profile abruptly drops off at the telomere-distal end. **g** Cartoons illustrate the effects of telomere association with the force-transmitting LINC complex on the alignment of chromosome ends.

chromosomes showed that the chromatin loop size was largely uniform along the chromosome length[4]. Our stage-resolved Hi-C data reconcile such a discrepancy. In early meiotic prophase I, we observed that transcriptionally active A compartment and inactive B compartment form domains preferentially consisting of short and long chromatin loops, respectively. The transcription-coupled variations in chromatin loop sizes suggest that the extension of meiotic chromatin loop is subject to the influence of transcription activity. While the exact mechanisms giving rise to these variations remain to be revealed, one possible explanation in the context of the loop extrusion model is that the high density of transcriptional machinery within the transcriptionally active regions could impede the movement of the loop extrusion complexes, resulting in slower rate of loop extrusion and smaller loops.

We note that the differences in chromatin loop organization for A and B compartment regions are not responsible for maintaining the active transcription during meiotic prophase. In fact, the disparity in loop sizes diminishes in later meiotic prophase I, when the highest levels of transcription occur[57,69]. Rather, we found that the regions consisting of smaller chromatin loops exhibit stronger inter-homolog interactions, providing evidence for its functional involvement in regulating homolog pairing and recombination.

The high abundance of COs in transcriptionally active regions has been largely attributed to the high frequency of DSBs, which are triggered by histone methyltransferase PRDM9[9,70]. Here we show that the genomic regions enriching for short chromatin loop sizes do not exhibit a higher density of DSB hotspots. However, these regions do exhibit higher ratios between COs and DSBs, implying that the structural properties of loop arrays could impact the probabilities of converting DSBs into COs. Together, our findings suggest a model in that the high transcriptional activity could influence chromatin loop sizes during early meiotic prophase I, creating a large-scale chromatin conformation that favors homolog recombination.

Our stage-resolved Hi-C also uncovered prominent alignment of chromosome ends among all homologous and non-homologous chromosomes during early meiotic prophase that extends over a substantial range. Recent large-scale single-sperm sequencing studies in mouse and human have revealed that the patterns of CO distribution are highly biased toward the chromosome ends[30,71]. In mouse, whereas COs are suppressed at centromere-proximal ends, the distribution of COs at the centromere-distal ends is largely concordant with the windows of elevated chromosome end alignment, implicating the regulation of chromosome conformation and spatial proximity between chromosome ends as a contributing mechanism for the modulation of CO occurrence.

We attributed the prominent chromosome end alignment to the association between telomeres and force-transmitting LINC complex using a mouse model carrying a single point mutation in SUN1 that disrupts the interaction between SUN1 and telomere. LINC complex plays an essential role during meiosis by enabling rapid chromosome movement, which is thought to facilitate homolog pairing and synapsis and resolve chromosome entanglement[28,72,73]. We found that, while the nuclear envelope association of telomeres was only partially affected by the SUN1 point mutation, the extensive alignment among chromosome ends was nearly completely abrogated. Our findings thus suggest that the LINC complex exerts a profound influence on the conformation of chromosome ends. While the LINC complex does not affect chromatin loop size at chromosome ends, it remains possible that the force transmitted via the LINC complex may modulate the stiffness, orientation, or physical length of the chromosome axis, hence allowing the chromosome ends to achieve better alignment over a considerable range. The precise nature of such force-induced conformational changes at chromosome ends would be a subject of interest for our future investigation.

In summary, our study provides rich resources and insights for understanding the spatial organization and functions of meiotic chromosomes. In particular, our data suggest transcriptional activity and mechanical force could modulate higher-order chromosome architecture during meiosis to regulate homolog pairing and alignment. Future functional studies by perturbing specific molecular players involved in these processes would not only further our understanding of the fundamental principles of meiotic processes but also open avenues for investigating how mammalian gametogenesis can be affected by intrinsic and extrinsic variables.

## Methods

**Animals**. C57BL/6J (B6, Shanghai Laboratory, Animal Research Center) male mice were housed under controlled environmental conditions with free access to water and food, with constant ambient temperature ($22 \pm 2\,°C$) and humidity ($55 \pm 10\%$), and an alternating 12 h light/dark cycle. C57BL/6J SUN1 W151R knock-in mice were generated by oocyte injection of CRISPR-Cas9 as described previously[74] and housed under the same conditions as the wild-type mice. Experimental protocols were approved by the regional ethical committee of the National Center for Protein Science Shanghai. Every effort was made to minimize and refine the experiments to avoid animal suffering.

**Spermatogenesis synchronization**. C57BL/6J mice were synchronized for isolation of leptotene spermatocytes as previously described[40]. Briefly, 2 dpp mice were pipette-fed 100 µg/g body weight WIN 18,446, which is suspended in 1% gum tragacanth, for 7 consecutive days. At 9 dpp, these mice were given a subcutaneous injection of 200 µg of RA. Seven days after RA injection, the testes of these mice were collected for leptotene isolation by FACS.

**Isolation of mouse spermatocytes by FACS**. Murine spermatocytes were isolated by FACS as described previously with modifications[75,76]. Wild-type or mutant mice of different ages were used for isolation of spermatocytes at different stages: 7–11-day-old mice were used for isolating Sertoli, spermatogonia, and preleptotene cells; 2-week-old mice for zygotene cells; 3–5-week-old mice for pachytene, diplotene, and MII; and 1–2-month-old SUN1 W151R mutant mice for zygotene-like spermatocytes (Supplementary Table 1). Decapsulated testes were digested with Collagenase IV (1 mg/ml)/DNase I (5 µg/ml) in Dulbecco's modified Eagle's medium (DMEM) at 37 °C for about 15 min until the appearance of dispersed tubules. The tubules were then collected and digested with 0.025% trypsin at 37 °C for 15 min. During digestion, the samples were gently pipetted every 5 min. Digestion was stopped by adding 4 volumes of DMEM with 10% FBS. Cells were collected by centrifugation, resuspended in complete DMEM, and filtered using a 70 µm strainer. Cells were then stained with Hoechst 33342 (5 µg/ml)/DNase I (5 µg/ml) at 37 °C for 1 h with gentle agitation every 10 min. Following staining, the cells were pelleted, resuspended in complete DMEM with propidium iodide (PI; 5 µg/ml)/Hoechst 33342/DNase I, and filtered through a 40 µm strainer. The cells were sorted on BD Influx (BD) cell sorter with the BD FACS™ Software (v1.2.0.142) to obtain different cell types.

We purified spermatocytes at different meiotic prophase I substages by combining stringent gating strategies with the use of mice at different ages. Debris was excluded based on the forward scatter and side scatter (SSC) parameters and intact cells were defied as Population 1 (P1), afterward trigger pulse width parameter was applied to the P1 to exclude cytoadherent cells and the remaining single cells were defined as Population 2 (P2). PI-negative population was defined as living cells and derived from P2, defined as Population 3 (P3). Different cell types from P3 can be distinguished by the fluorescent profiles of two emission channels of Hoechst (Hoechst Blue and Hoechst Red). Testicular cells from 3- to 5-week-old mice exhibit a consistent Hoechst Blue/Hoechst Red fluorescent profile on which the Sertoli, spermatogonia, preleptotene, L/Z, P/D, and MII cells each occupy a different region. While the isolation of mixed L/Z and P/D populations has been achieved previously[34], the separation of leptotene from zygotene and pachytene from diplotene spermatocytes was technically challenging. By using smaller gates set toward the left end and the right end of the P/D zone, we were able to obtain pachytene and diplotene at high purify from 3- to 5-week-old mice, albeit at a cost of a lower yield of cells. On the other hand, the leptotene and zygotene cells isolated using the same strategy suffered from contamination of pachytene cells. We found that this problem can be mitigated by using mice at a younger age. When using 2-week-old mice and a small gate set at the right side of the L/Z zone, the zygotene cells can be purified. However, leptotene cells isolated using a similarly small window set at the left side of the L/Z zone still contained considerable zygotene cell contamination. We further overcame this problem by

performing FACS using the testicular cells from synchronized, 16-day-old mice, in which the fraction of zygotene-stage cells was much smaller. Finally, we found that the use of 7–11-day-old mice led to the isolation of preleptotene-stage spermatocytes at the highest purity. Testicular cells from one or two 3–5-week-old mice or 5–8 mice at younger ages were pooled together and used for FACS sorting in each experiment (see Supplementary Table 1). The viability and purity of cells isolated from each experiment were further assessed by performing immunofluorescence (IF) on a fraction of cells (see the "Quality control of spermatocytes by chromosome spreading and IF" section below). The rest of the cells were fixed immediately using 1% v/v formaldehyde and saved for Hi-C library preparation (see the "Hi-C library construction" section below). Typically, we were able to obtain ~50,000–200,000 cells for the desired stage from each experiment. A total of 223 C57BL/6J mice were used to isolate sufficient cells for generating the Hi-C datasets in this study.

**Quality control of spermatocytes by chromosome spreading and IF**. After each FACS experiment, a 10 µl aliquot of sorted cells was mixed with 10 µl 0.4% trypan blue, and the cell viability and concentration were assessed using an Invitrogen Countess II cell counter. Chromosome spreading and IF were subsequently performed to assess the purity of meiotic prophase I spermatocytes at different stages as previously described[76]. Briefly, about 15,000–20,000 sorted spermatocytes were mixed with 1 ml hypotonic buffer (30 mM Tris-Cl, 50 mM sucrose, 17 mM sodium citrate dehydrate, 5 mM EDTA in water) on ice for 3 h. Cells were pelleted, resuspended, and applied to poly-lysine-coated coverslips pretreated with 1% paraformaldehyde (PFA) and 0.1% Triton X-100, pH 9.2. Coverslips with cells were incubated in a humid chamber overnight, rinsed with 0.4% Photo-Flo 200 twice for 2 min, and dried at room temperature (RT). Coverslips were blocked in 5% bovine serum albumin (BSA) for 1 h and subsequently incubated with primary antibody at 1:200 dilution (anti-SYCP3 (Abcam, ab15093) and anti-phospho-Histone γH2AX (Novus, NB100-384)) for 2 h at RT. After washing with PBST buffer, coverslips were incubated with secondary antibodies at 1:1000 dilution (goat anti-rabbit secondary antibody, DyLight550 (Thermo, 84541) and goat anti-mouse secondary antibody, DyLight488 (Thermo, 35502)) for 1 h, washed with PBST containing 4,6-diamidino-2-phenylindole (DAPI; Sigma, D9542), mounted to the slides with FluorSave™ Reagent (Merck, 345789). For Sertoli and spermatogonia cells, IF was performed without using the chromosome-spreading procedure. Sertoli and spermatogonia cells were centrifuged directly onto slides with Cytospin and stained with anti-FSHR (Abcam, ab113421, 1:200 dilution) and anti-DMRT1 (Santa Cruz, sc-377167, 1:200 dilution) as described above. Imaging was performed using a Zeiss LSM 880 confocal microscope with a ×63, 1.40 NA oil lens. Confocal images were acquired with the Zeiss Zen software (2012 S4).

We categorized the meiotic prophase I spermatocytes remained on the slides using the following criteria based on patterns of DAPI, SYCP3, and γH2AX distribution[76]: Cells were categorized as preleptotene spermatocytes if they exhibited weak, diffused, or punctate SYCP3 but no obvious stretches of SYCP3 signals. Cells showing short-to-long SYCP3 stretches that do not span over the entire length of the chromosome were categorized as leptotene cells. Cells were categorized as zygotene cells if they exhibited long SYCP3 fibers that indicate the formation of chromosome axes, as well as strong γH2AX signals, but did not exhibit the XY sex body that is intensely labeled by γH2AX. Pachytene cells could be identified based on the "stubby" SYCP3 signals that indicate the fully synapsed tetrad chromosomes, as well as the appearance of the γH2AX-labeled sex body. Finally, cells were categorized as diplotene if they exhibited partially separated chromosome axes. After the quality assessment, the isolated cell populations that were of <80% viability or 78% purity were discarded and not used for downstream experiments. The cells that met our selection criteria were further pooled into two batches for Hi-C library construction. The final purity for the pooled cells was >83% for leptotene and zygotene-stage spermatocytes and >90% for all the other stages.

**IF combined with FISH**. To examine the three-dimensional (3D) organization of chromosomes and localization of telomeres in the isolated cells, IF alone or IF combined with FISH (IF-FISH) was performed on spermatocytes as previously described[77]. The isolated cells were centrifuged directly onto slides with Cytospin without being treated with the hypotonic buffer. IF process was carried out as described above. After immunostaining with primary and secondary antibodies, slides were washed and fixed with 4% PFA for 10 min, rinsed with PBS, and dehydrated in 70, 85, and 100% ethanol. Samples were then dried at RT, denatured at 85 °C for 5 min in the presence of Cy3-labeled (CCCTAA)₄ PNA probes (TelC) (Panagene), and cooled to 37 °C at a rate of 0.1 °C/s in a thermocycler. Following hybridization, slides were rinsed with hybridization buffer, washed twice with 2×SSC buffer and once with PBS supplemented with 1 µg/ml DAPI. After being dried at RT, slides were mounted with FluorSave™ reagent. Microscopy imaging was performed by a Leica TCS SP8 STED 3× with a ×63, 1.40 NA oil lens, and images were captured with Leica application suite X (v3.7.0).

**Hi-C library construction**. Hi-C libraries of different cell types were prepared using a previously described in situ Hi-C protocol with minor modifications[44]. In all, 0.1–1 million cells were fixed with 920 µl 1% v/v formaldehyde in 1× HBSS at

RT for 10 min with mixing. Also, 2.5 M glycine was added to the cells to make the final concentration of glycine 0.2 M. Cells were incubated at RT for 5 min with mixing and on ice for at least 15 min to quench crosslinking. The fixed cells were centrifuged at 3000 × g for 10 min at 4 °C. The supernatant was discarded and cell pellets were resuspended with 100 µl ice-cold Hi-C lysis buffer (10 mM Tris-HCl pH 8.0, 10 mM NaCl, 0.2% Igepal CA630) with protease inhibitors. Samples were incubated on ice for 15 min and centrifuged for 5 min at 3000 × g at 4 °C to isolate nuclei. The nuclei pellets were washed once with 100 µl 1× NEB buffer 3.1, resuspended in 2 µl 10× NEB buffer 3.1, 14 µl ddH₂O, and 4 µl of 0.5% sodium dodecyl sulfate (SDS) and then incubated at 65 °C for 5 min to open up chromatin. In all, 2.2 µl of 10% Triton X-100 (Sigma, 93443) was added to the samples, and samples were incubated at 37 °C for 15 min to quench SDS. To digest chromatin, 0.5 µl of 10× NEB 3.1 buffer, 1.5 µl ddH₂O, and 1 µl (50 U) of DpnII restriction were added into the samples to bring total reaction volume to ~25 µl. Digestion was performed overnight at 37 °C on a thermomixer with interval shaking (shake at 950 rpm for 10 s with 5 min intervals). Following digestion, samples were incubated at 65 °C for 20 min to inactivate the restriction enzyme, then cooled to RT on ice. To fill the restriction overhangs, 6 µl of fill-in master mix (3.75 µl of 0.4 mM biotin-14-dATP (Life Technologies, 19524-016), 0.15 µl of 10 mM dCTP/dGTP/dTTP, 1 µl of 5 U/µl Klenow fragment (3'→5' exo-), 0.6 µl 10× NEB 3.1, 0.2 µl Milli-Q water) was added to the reaction and samples were incubated at 37 °C for 1 h in thermomixer (shake at 950 rpm for 10 s with 5 min intervals). In all, 90 µl of ligation master mix (49.8 µl of water, 24 µl of 5× T4 DNA ligase buffer (Invitrogen), 10 µl of 10% Triton X-100, 1.2 µl of 10 mg/ml BSA (100× BSA), 5 µl of 1 U/µl T4 DNA Ligase (Invitrogen)) was then added and samples were mixed by inverting and incubated at 16 °C for 4 h. After ligation, samples were centrifuged for 5 min at 3000 × g. The pellets were washed with 100 µl 1× NEB buffer 3 and resuspended with 50 µl 1× NEB buffer. To reverse crosslinking, 3 µl 20 mg/ml proteinase K (NEB, P8102) and 5 µl of 10% SDS were added into the samples and then incubated at 65 °C for 2 h. Phenol–chloroform extraction and ethanol precipitation were performed. The purified DNA was dissolved in 130 µl 1× Tris buffer and sheared by Covaris M220 (peak incident power: 50 W; duty factor: 20%; cycles per burst: 200; treatment time: 65 s). To minimize sample loss, size selection was not performed after shearing. End repair of DNA fragments and ligation with Illumina Truseq adapters was performed using End Repair/dA-Tailing Module (NEB, E7442L) and Ligation Module (NEB, E7445L) according to product manuals. Following adapter ligation, the total volume of DNA samples was brought to 300 µl with Milli-Q H₂O. To purify biotinylated DNA fragments, 50 µl Dynabeads MyOne Streptavidin T1 beads (Life technologies, 65602) were washed twice with 100 µl 1× Tween washing buffer (TWB, 5 mM Tris-HCl (pH 7.5); 0.5 mM EDTA; 1 M NaCl; 0.05% Tween 20) and then resuspended in 300 µl 2× binding buffer (10 mM Tris-HCl (pH 7.5); 1 mM EDTA; 2 M NaCl) before being added to DNA samples. The samples were incubated at RT for 15 min with rotation to bind biotinylated DNA to the streptavidin beads. Beads were washed twice with 100 µl 1× TWB on a Thermomixer at 55 °C for 2 min with mixing and resuspended in 50 µl Milli-Q water. Samples were amplified with Phusion High-Fidelity DNA Polymerase (Thermo, F-530L) for 4–7 cycles. For size selection, 0.5× volumes of AMPure XP beads (Beckman Coulter, A63881) prewarmed to RT were added to the PCR products, and samples were incubated at RT for 5 min. Samples were then separated on a magnet. The clear solution was mixed with another 0.5× volumes of AMPure XP beads by pipetting and incubated at RT for 5 min. Samples were again separated on a magnet and the clear solution was discarded. Beads were washed once with 700 µl freshly made 70% ethanol without mixing. After removing ethanol, tubes containing beads were left on the magnet for 5 min with lids open to allow the remaining ethanol to evaporate completely. DNA was eluted from beads with 25–50 µl of 1× Tris buffer. Adapters and primers used in Hi-C library construction are listed in Supplementary Table 4. All Hi-C libraries were sequenced on Illumina Nova Seq 6000 platform at PE150 mode.

**CUT&Tag library construction**. Sertoli cells, preleptotene spermatocytes, mixed L/Z spermatocytes, and mixed P/D spermatocytes were purified from the testes of two 3-week-old mice using FACS as described in Vara et al.[34]. A less stringent gating strategy than the strategies described above for Hi-C library construction was used; as a result, the leptotene and zygotene cells, as well as the pachytene and diplotene cells, were not separated from each other. CUT&Tag libraries were generated as previously described[78] with the Hyperactive In-Situ ChIP Library Prep Kit for Illumina Kit (Vazyme Biotech, TD901). Briefly, 50,000 cells for each CUT&Tag library were incubated with 10 µl balanced concanavalin A-coated magnetic beads at RT for 10 min. The bead-bound cells were resuspended with 50 µl of antibody buffer. In all, 1 µg rabbit monoclonal anti-CTCF antibody (Abcam, ab128873) or rabbit IgG (Beyotime, A7016) was then added to the bead-bound cells. After 2-h incubation at RT, the primary antibody was discarded carefully and 0.5 µl goat anti-rabbit IgG (Vazyme, Ab206-10-AA) diluted with 50 µl Dig-wash buffer was added to the cells. The cells were incubated at RT for 1 h. Afterward, samples were washed gently with 800 µl Dig-wash buffer, and 0.58 µl pG–Tn5 together with 100 µl Dig-300 buffer was added to the samples. After 1-h incubation at RT, samples were washed gently with 800 µl Dig-300 buffer. In all, 300 µl tagmentation buffer was then added to each sample and the samples were incubated at 37 °C for 1 h. The interactions were quenched by adding 10 µl 0.5 M EDTA, 3 µl 10% SDS, and 2.5 µl 20 mg/ml Proteinase K and incubating the samples

at 50 °C for 1 h. Phenol–chloroform and ethanol precipitation, as well as PCR and size selection, were performed as described in Hi-C library construction. All CUT&Tag libraries were sequenced on Illumina Nova Seq 6000 platform at PE150 mode subsequently.

**RNA-seq library construction**. Preleptotene, leptotene, and zygotene spermatocytes were purified and assessed from 10-day-old unsynchronized mice, 16-day-old synchronized mice, and 2-week-old unsynchronized mice, respectively, using the same stringent gating strategies as for constructing Hi-C libraries. The viability and purity of isolated cells are summarized in Supplementary Table 2. For each stage, two independent biological samples were isolated and used for RNA-seq library construction. Total RNA was extracted from ~50,000 purified spermatocytes by the RNeasy Micro Kit (Qiagen, 74004) according to the manufacturer's instructions. The cDNA was synthesized and amplified using the Single Cell Full-Length mRNA-Amplification Kit (Vazyme, N712). The RNA-seq libraries were constructed from the amplified cDNA using the TruePrep DNA Library Prep Kit V2 for Illumina® (Vazyme, TD503). The PCR and size selection of RNA-seq libraries were conducted as described in Hi-C library construction. The RNA-seq libraries were sequenced on Illumina Nova Seq 6000 platform at PE150 mode.

**Hi-C data processing**

*Mapping, binning, and heatmap generation*. Hi-C reads were mapped to the mm10 genome (https://hgdownload.soe.ucsc.edu/goldenPath/archive/mm10/) and filtered using the hiclib pipeline (https://bitbucket.org/mirnylab/hiclib) as described previously[52]. Mapping statistics are summarized in Supplementary Table 3. The mapped Hi-C fragments of replicates were combined and converted into cooler format using the cooler package[79] (https://github.com/mirnylab/cooler, v0.8.5). The cool files were balanced and binned into multiple resolutions (10 kb, 20 kb, 50 kb, 100 kb, 500 kb, and 1 Mb) for subsequent downstream processing. The *cis* and *trans* expected contact probability at different genomic separation was calculated using the cooltools package (https://github.com/mirnylab/cooltools, v0.2.0). The observed heatmaps and observed/expected heatmaps at 10 kb, 50 kb, or 1 Mb resolutions in the figures were generated using scripts derived from the cooltools package.

*Evaluation of reproducibility of Hi-C replicates*. Pearson correlation coefficients between Hi-C replicates for each stage or between datasets of different stages were calculated using cool files binned at 500 kb and the HiCRep package[80] (https://github.com/TaoYang-dev/hicrep, v1.11.0). The optimal smoothing parameter was set as 1, which was determined by the function htrain, and the lower and upper bounds of the genomic distance between interaction loci were set to 0 and 5,000,000 as recommended. Reproducibility scores for the whole genome of each sample were generated as the mean value of all chromosomes. Pearson correlation coefficients were also calculated between the pairwise combination of the insulation profiles at 10 kb resolution for Hi-C datasets (see "TAD analysis").

*P(s) curves and loop size derivation*. Contact probability ($P(s)$) curves were computed from the cool files binned at 10 kb resolution. The linear genomic separations were divided into logarithmic bins with a factor of 1.12. The $P(s)$ values were calculated by averaging the chromatin interaction frequencies within each log-spaced bin for the combined autosomes as well as each autosome and the X chromosome. The $P(s)$ values were plotted in log space versus the genomic distances using custom R scripts to generate $P(s)$ curves. To derive the average chromatin loop sizes, the log-space slopes for $P(s)$ curves were calculated and plotted against the log distances ($\log(s)$) after loess smoothening[46]. In these $P(s)$ derivative plots, the genomic distances showing maxima in $P(s)$ slope values correspond to the average lengths of chromatin loops in that dataset. In Fig. 2a–d, the $P(s)$ and $P(s)$ derivative plots were generated using the chromatin interaction frequencies for all autosomes. In Fig. 4d, e, only the interactions within A or B compartment regions >2 Mb were used.

*Chromatin loop pileup*. Pileup analysis of chromatin loops was performed on Hi-C datasets binned at 10 kb resolution using the coolpuppy package[81] (https://github.com/Phlya/coolpuppy, v0.9.5). Interactions among specific sites that are at a minimal distance of 500 kb and a maximal distance of 2 Mb were used to generate pileups. The pileups were normalized to randomly shifted control regions (10 controls per region of interest). The pileup heatmaps were plotted using matplotlib.

*TAD analysis*. Insulation analysis and TAD calling were performed on cooler matrices of combined Hi-C replicates binned at 10 kb resolution using the diamond-insulation utility from the cooltools. Insulation indices were calculated by averaging signals within a 500 kb insulation square. To assess the reproducibility of Hi-C datasets, Pearson correlation coefficients were calculated between the pairwise combination of the insulation profiles. TAD boundaries were identified using the default parameter. A noise threshold of 0.1 was further used to select the most prominent TAD boundaries. Insulation profiles were plotted using custom scripts in R. The pileups of 500 kb genomic regions flanking the TAD boundaries were performed on Hi-C datasets binned at 10 kb resolution using coolpup.py under the

–local mode and normalized using the expected interaction probabilities at each genomic distance.

*Analysis of chromatin interactions around DSBs*. The positions of DSB hotspots were obtained from the previously published DMC1 ChIP-Seq data[56] and converted into 10 kb genomic bins. The positions of CO hotspots were obtained from the single sperm DNA sequencing data[30]. The 10 kb genomic bins containing DSBs were classified as CO-DSBs if they also contain CO hotspots and as NCO-DSBs if they are >100 kb away from the nearest CO hotspots. Genomic bins generated by randomizing the positions of all CO-containing bins using the bedtools shuffle utility were used as the negative controls. Pileup heatmaps for 2 Mb genomic regions centered at each class of genomic bins were generated from the cool files using the coolpuppy package under the –local and normalized using the expected interaction probabilities at each genomic distance. The total balanced interactions within each 2 Mb genomic region or along the center line of each region were calculated using the fetch function from the cooltools package.

*Compartment analysis*. Eigenvalue decomposition was performed on cooler matrices binned at 10 kb resolution using the call-compartments utility from the cooltools package. The first eigenvector profiles for specified chromosomes or genomic regions were generated using custom scripts in R. 10 kb genomic bins with positive first eigenvector values are within A compartment, while bins with negative first eigenvector values are within B compartment. The fractions of genomic bins that changed compartment identity were quantified using custom scripts in R. To assess the changes in compartment strength during meiosis, saddle plots were generated using the compute-saddle utility from the cooltools package.

*Inter-chromosome interaction analysis*. Cooler matrices binned at 10 kb resolution were used to analyze the inter-chromosome interactions between chromosome ends. We consider the 5 Mb regions at either the centromere-proximal end (CEN) or the centromere-distal end (TEL) as sub-telomeric regions. The average chromatin interaction frequency in each 10 kb bin was calculated for every pairwise combination of sub-telomeric regions. The polarity of chromosome end interactions was assessed by calculating (CEN-CEN + TEL-TEL)/(2 × CEN-TEL) for each of the 171 pairwise combinations of autosomes. Boxplots depicting changes in chromosome end association during meiosis were generated using the ggplot2 package in R. To assess the alignment between different chromosomes, *trans* observed interaction frequencies for each of the 171 pairwise combinations of autosomes were obtained from cooler matrices binned at 50 kb resolution. The *trans* observed/expected matrices were generated by dividing the observed matrices with the *trans* expected contact probabilities calculated with the compute-expected utility from the cooltools package. The 171 *trans* observed/expected matrices were then rescaled to 500 bins × 500 bins by interpolation, averaged, and plotted using matplotlib in python. To assess the extent of chromosome alignment along chromosome length, we calculated the diagonal signals of the averaged matrices by sliding a 21 bins × 21 bins square centered at each diagonal bin. The average signal within the square was then assigned to the diagonal bin. The diagonal signal profiles were loess smoothed using a span of 1/20 and least-squares fitting and plotted in R.

*Quantification of inter-homolog alignment using Patel et al. Hi-C data*. Inter-homolog Hi-C data for zygotene-stage spermatocytes from the Patel et al.[32] study were downloaded from NCBI Gene Expression Omnibus (GEO) (accession: GSE122622). The GSE122622_zygonema_interhomolog.hic file was then converted to cooler format using the hic2cool package (https://github.com/4dn-dcic/hic2cool, v0.8.3) and binned at 50 kb. To quantify the inter-homolog alignment scores for each 50 kb genomic interval, we calculated the mean signal within 21 bins × 21 bins square centered at each diagonal bin on the inter-homolog interaction heatmaps. These values were then normalized using the mean inter-homolog interaction frequency per 50 kb bin for each pair of homologous chromosomes. Comparison of inter-homolog alignment scores for different sets of genomic regions was performed using custom scripts in R.

**Quantification of DSB and CO density**. To examine the correlation between large-scale chromatin organization and the distribution of DSB and CO events, we used the DSB hotspots identified based on the DMC1 ChIP-Seq from two independent studies[56,58], which we named as the Davies DSBs and the Smagulova DSBs, respectively, as well as the CO hotspots identified based on the single sperm DNA sequencing[30]. We counted the number of DSB or CO hotspots within each A compartment or B compartment region >1 Mb identified in the zygotene stage and calculated the number of DSB or CO hotspots per Mbp for each A/B compartment region. We further derived the ratio between the COs and DSBs (COs/DSBs) using both DSB datasets. The average balanced interaction probability at 500 kb for each A/B compartment region >1 Mb was calculated using the expected.diagsum function from the cooltools package. The relationship between the 500 kb interaction probability and the Davies DSB density, the Smagulova DSB density, the CO density, the COs/Davies DSBs ratio, and the COs/Smagulova DSBs ratio was examined and plotted in R.

**RNA-seq data processing**. All RNA-seq raw data were trimmed to remove low-quality reads and adapters with Trim Galore (v0.6.5). The clean reads were then mapped to the mouse reference genome (mm10) using the STAR aligner[82] (v2.7.3a) with default parameters and the annotated genes were counted by HTSeq[83] (v0.12.4). The raw counts files were normalized using the DESeq2 (v1.22.2) package and heatmaps in Supplementary Fig. 3 were generated using the ggplot2 (v3.3.0) and pheatmap (v1.0.12) packages in R. Coverage tracks at 10 bp (Supplementary Fig. 3) and 100 kb (Supplementary Fig. 14) bin sizes were generated using the bamCoverage (v3.3.1) utility from the deeptools (v3.3.1) package[84] with the parameter "-normalizeUsing RPKM."

**Analysis of Vara et al. ChIP-Seq data**. P/D-stage CTCF and REC8 ChIP-Seq data[34] were downloaded from GEO (accession: GSE132054). In this study, less stringent FACS gating parameters were used to isolate the mixed populations of P/D spermatocytes. The downloaded fastq files were aligned to the mm10 genome using bowtie2[85] (v2.4.2) with the parameters "–very-sensitive -X 2000." Peaks were called using the MACS2[86] (v2.1.2) callpeak function with input data as control. The peaks occupied by both CTCF and REC8 peaks were identified using the bedtools intersect utility. The quantification of ChIP-Seq signals on different sets of peaks and the generation of boxplots were performed using custom scripts in R. The CTCF and REC8 occupancy at the peaks called using macs2 was further visualized using the computeMatrix function with the parameters "scale-regions -b 2000 -a 2000 -m 2000" and the plotHeatmap function from the deeptools package[84]. Genomic regions generated by randomizing the 17,613 CTCF peaks or the 13,561 REC8 peaks using the bedtools shuffle utility were used as the negative controls for visualizing CTCF and REC8 ChIP-Seq data, respectively.

**CUT&Tag data processing**. The CTCF CUT&Tag data processing and analysis were conducted as described before[78] with minimal changes. Low-quality reads and adapters were trimmed by Trim Galore (v0.6.5). The clean reads were mapped to the mm10 genome with bowtie2 (v2.4.2)[85] and duplicates were removed with Picard (http://broadinstitute.github.io/picard/, v2.18.29). The CTCF occupancy at different categories of CTCF or REC8 peaks for each CUT&Tag sample was visualized using the deeptools package as described above.

**Reporting summary**. Further information on research design is available in the Nature Research Reporting Summary linked to this article.

## Data availability

The data that support this study are available from the corresponding authors upon reasonable request. All raw data of high-throughput sequencing and processed files for wild-type cells used in this study have been deposited in the National Center for Biotechnology Information (NCBI) Gene Expression Omnibus (GEO) under the accession codes GSE155638 and GSE155967. The Hi-C data for SUN1 W151R mutant zygotene spermatocytes have been deposited in GEO under the accession code GSE155142. The following publicly available datasets used in the manuscript were downloaded using GEO platform: inter-homolog Hi-C data for zygotene-stage spermatocytes (GSE122622); P/D-stage CTCF and REC8 ChIP-Seq data (GSE132054). Source data are provided with this paper.

## Code availability

Custom scripts used in this study are publicly available at https://github.com/bianlab-hub/zuo_ncomms_2021/tree/Hi-C_data_analysis with https://doi.org/10.5281/zenodo.5282919[87].

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

## Acknowledgements

The authors thank the Flow Cytometry Lab and Bioimaging Facility in Shanghai Institute of Precision Medicine for help with experiments. We thank S. Yu and A.C. Chang for helpful discussions. This work was supported by the National Key Research and Development Program of China (2018YFC1004703 to Q.B. and 2018YFA0107004 to M.L.) and the National Natural Science Foundation of China (31870749 to G.C., 31801056 and 31970585 to Q.B., U1632267 to M.L., 31971137 to C.H.). Q.B. is also supported by The Program for Professor of Special Appointment (Eastern Scholar) at Shanghai Institutions of Higher Learning (TP2018044).

## Author contributions

Q.B. and M.L. conceived and designed research; W.Z., G.C., Z.G., S.L., Y.C., C.H. and J.C. performed experiments; Q.B. and W.Z. analyzed data; Q.B., M.L. and Z.C. supervised experiments and data analysis; Q.B. wrote the manuscript; M.L., W.Z. and C.H. edited the manuscript.

## Competing interests

The authors declare no competing interests.
