## [Peer Review File · Nature Communications]

REVIEWER COMMENTS

Reviewer #1 (Remarks to the Author):

Overview: Here the authors present a detailed analysis of mouse meiotic chromosome conformation as measured by HiC. Key analytical findings are that compartment A and B domains may have differing compaction rates, perhaps suggesting differential loop size, and that this varies during the meiotic stages. The authors also demonstrate significant interchromosomal interaction of subtelomeric regions (most likely detection of a bouquet-like stage) that is largely abrogated by Sun1 mutation. The data appear high quality, are well presented, and the manuscript is clearly written. However, in a number of places, data are over-interpreted, and measurable statistical differences need to not be confused with something having a meaningful biological effect/difference.

Notably, a number of relatively high-profile meiotic HiC studies (in various organisms) were published in the past year or so in: NSMB, Molecular Cell, Cell Reports, and Nature Communications. The authors do adequately reference these studies, although I have some concerns that the new datasets and analyses that are presented in this manuscript are relatively incremental in their impact compared to those prior works. Nevertheless, overall, this is a well-constructed study, that requires, mostly, only text changes to address the concerns and questions that I have raised.

General. To maximise accessibility, I would suggest that the authors do not use the abbreviations P, Z, L, preL etc for the various stages. These seem unnecessary, frequently making the reading of the text challenging to follow.

General: For all HiC plots, please indicate in the image or legend the bin size that is being plotted/analysed.

Specific comments and suggestions:

Line 90-95. Whilst it is true that many of the prior studies included pachytene-stage samples, Patel 2019 included zygonema stage and Vara 2019 included a Leptonema/Zygonema sample. Please revise text to ensure that prior literature is represented accurately.

Line 107. Similarly, it is an unnecessary overstatement to say that these data are "highly demanded" when prior work already exists. The authors, here, present nice new data and observations that build on prior findings. There is no need to overstate things. Please recognise the prior work more thoughtfully.

Section 130-142. A much more detailed description of the criteria used to isolate and validate the specific stage-resolved populations is required. This is a critical part of the study, yet almost no detail is presented. Specifically, what aspects of the FACS data and microscopy enable the designation of specific staged populations?

Line 148. Where does Fig 1c demonstrate telomere clustering at Zygotene? For this to be informative, a zoom in and/or aggregation, and comparison with other stages is required. In general, it is very unclear what Fig 1c is meant to demonstrate. In particular the text in the legend describing Fig 1c: "...shows dynamic reorganization of 3D genome architecture during meiosis" is not supported by the content of this figure. Please clarify and/or revise.

Line 149. The correlation coefficients are almost universally high irrespective of whether the data being compared are replicates or non-replicates (this is especially true of the Pre-Leptotene through to Diplotene samples (S1b) where all values are >0.95). As such, I find it hard to be persuaded that this analysis demonstrates that the replicates are highly reproducible. For example, if (all) the samples just so happened to be full of homogenous noise (once binned at 50 kb), the data would also be correlated...but this would not in any way indicate that the samples are high quality. Thus, perhaps there is a better way to demonstrate sample similarity than such a crude global correlation? For example, perhaps there are stage-specific visible patterns that can be discerned in replicates but not in non-replicate samples.

Line 150. Please elaborate on how Fig S2b demonstrates that the cell stages are distinct from one another? Is the difference between correlation coefficients of 0.98 and 0.96 really meaningful? It suggests that there is very little difference in HiC contact maps between stages, and/or that the staged samples are not well isolated from one another temporally. Which is it?

Line 152-153. Without more convincing evidence, this comes across as an overstatement.

Fig 2a-d and associated text (Lines 159-190). The curves are very similar and overlap in complex patterns. As such, please present separate curves for each replicate and/or some indication of the variance between the repeats. Adding this information will enable reader to appraise how much

variation there may exist between biological replicate samples of the same stage relative to different biological samples.

If we assume that there is a difference in compaction at different stages as measured by P(S) in panels 2b-d (but not on Chromosome X), it would be useful to determine and present whether this compaction difference arises in all autosomes, or only in some.

Lien 189. Please clarify in the text why Fig S1a is referred to here.

Line 189-190. Without additional experiments and observations, there is nothing yet presented in the manuscript that "reveal[s] that progressive loop extrusion occurs throughout the entire meiotic prophase I". The data *may* be consistent with this interpretation/model, but the data far from "reveal" it to be the case. There could easily be other mechanisms that alter chromosome conformation. In particular, there is not, to my knowledge, direct evidence yet of loop extrusion in meiotic cells.

In Fig S4a, please could the positions of the boundaries be more clearly indicated on the axes? The relevant tick marks are currently almost invisible. Moreover, it would be helpful if these plots were represented as fold changes compared to expectation (essentially a genome average contact matrix), since this will normalise for the expected interaction all-off that happens with increasing distance from the diagonal, and thus should emphasise any boundary features.

Line 201-203. And Fig S4c. How many of the boundaries overlapped in each stage? It is possible to present a Venn diagram of these data? i.e. Are the boundaries that are present in the L-D samples mostly the same? And are they a subset of the boundaries detected in the Sertoli etc cells, or completely different?

Line 206. It is not clear that the boundaries have "marked weakening" in PreL stage. Fig S4c seems to indicate weakening at the spermatogonia stage. Please clarify and/or revise.

Line 216-220. Please revise main text to make it clear that this analysis was performed using the pachytene stage data.

Line 219-220. Given that there are approximately three times as many REC8-only sites as REC8+CTCF sites, is it reasonable to conclude that, even despite lower peak ChIP signal, that the REC8-only sites are the "preferential binding sites"? i.e. It seems entirely possible that, globally, there is more REC8 at non-CTCF sites than at CTCF sites... Moreover, REC8 peaks (whether at CTCF or not) may represent only a tiny fraction of all the, potentially heterogeneously-bound, REC8.

Line 221-222. Please indicate the total number of CTCF peaks analysed in the two classes. Indeed, for clarity, perhaps the number of peaks can be added to both Fig 2e and 2f.

Fig 2g-h, For clarity, please label the cell stages that were analysed in the title of these figures (as in 2e-f), rather than in the Y-axis.

Line 231-234. Please revise this text, since the conclusion does not seem to make sense. If the insulation score goes up more, how/why does it indicate greater *loss* of insulation ability?

Secondly, whilst the difference between the red and orange distributions is significant, this is likely to be only because there is so much data. i.e. My point is this: Quantitatively, is there really any substantial difference?

Line 252-253. Please can the authors clarify in the text what they mean by "more dynamic"? Do the authors infer greater heterogeneity in loop positions to mean greater dynamism? Is there evidence to support this? Loops in interphase could be well-positioned across a population, yet highly dynamic, for example.

Fig S5a. Perhaps the y-axis should be relabelled as percentage of maximum (as observed in PreL)? At present it is rather odd to have $A+B=200\%$. What are the absolute proportions of compartment A and B (i.e. is A or B more frequent?) This information appears to be hidden for no obvious reason.

Lines 270-272. Fig S5a. It appears that it is specifically B that reduces during Zygotene to Diplotene transition. Perhaps this should be commented upon?

Line 270-271. Please elaborate in the text on how Fig 3a demonstrates gradual attenuation of compartmentalization from PreL to D. PreL looks clearly different (but please explain in what manner). The other stages look more or less indistinguishable in this representation.

Line 271-272. Please elaborate in the text how Fig S5b supports the statement that: "regions of different transcriptional status rarely intermingle with each other".

Line 273-274. Where is the evidence for this statement? The boundaries between A and B are not discernible in the raw data in Fig S5b L to D stages.

Lines 276 to 282 and Fig 3b-f. This is a very nice set of analyses. Very intriguing!

Line 313-314. Could the authors elaborate in the text why they think more DNA near the chromosome axis would aid homolog recognition? This isn't obvious to me.

Line 328 and Figure 4c. As mentioned above, whilst highly statistically significant (due to lots of data), the actual absolute difference here in "alignment precision score" between A and B is extremely small. I would caution the authors not to over-interpret the significance of such tiny differences. i.e. the important point here is not the statistical significance, but the effect size (which is actually tiny).

I have a similar concern about Fig 4e. These data distributions, whilst apparently statistically different, are also incredibly similar. So does this really mean anything?

Line 361. and Line 362-364. Where in Fig S1a is the data to back up this statement that <5% of the cells in Zygotene exhibited bouquet conformation? Importantly, consider how were the IF images prepared. Methods indicate that chromosomes were spread. Does this maintain or does this disturb bouquet structure? I would suspect the latter... Without further evidence one way or the other, the conclusion(s) in this section need substantial revision.

Line 397-412. It is rather hard to appraise this section since so many of the observations that support the ideas are listed as "unpublished". If they are relevant to the logical flow of this manuscript should they not be included here?

Line 423-430. It should perhaps be mentioned that at the very tips of the chromosomes the observed interchromosomal interaction frequency is now *increased* in the Sun1 mutant. Thus contacts are both increased, yet persist of a shorter fraction of the chromosome length.

Line 448. Please revise the mention of dynamic loop extrusion in line with comments made above.

Reviewer #2 (Remarks to the Author):

In this work, Zuo and collaborators mapped the 3D genome architecture through spermatogenesis. Obtaining HiC maps from highly differentiated cells such as germ cells is quite a challenge so I acknowledge the effort to conduct the work presented here. Also, the paper is well written given the complexity of the data generated.

Given said that, my major concern relay on the novelty of the data presented. The 3D maps provided in the present manuscript are in fact patterns have already been reported in prior studies (Wang et al. 2019; Patel et al. 2019; Alavattam et al. 2019; Vara et al. 2019). It is true that pre-leptotene stage and Sertoli cells maps are new in the literature, but it is not clear how authors are able to differentiate among all cell types during prophase I. In fact, I have concerns regarding the purification process (see specific comments below).

Also, the description of the progressive increase of chromosome loop size throughout meiotic prophase I (lines 158-190) have also been reported previously, not only in mammals but also in yeast. The same occurs for interactions at telomeric regions (lines 337-348).

Later on the manuscript (lines 192-255), authors claim that anchoring chromatin loops by REC8 enables dynamic regulation of loop extrusion during meiosis, but based on the results presented it is not clear how the authors arrive to that conclusion. Also, authors talk about TADs and loops, but are they able to distinguish between these two genomic features based on their data?

In addition, the authors conclude that transcription-coupled variations in chromatin loop sizes during early meiotic prophase (lines 257-306). But it is not clear which transcription data are the authors considering for their analysis. While A/B compartments are defined based on eigenvector (what the authors are studying here) active transcription regions are based on RNA-seq data. Often both features are correlated but this needs to be proven.

Authors also state that '... our analyses reveal an extensive correlation between transcription-coupled variations in chromatin loop size and the extent and precision of homolog alignment, implying that chromatin loop sizes may contribute to the modulation of meiotic recombination landscape' (lines 32-335). I think this is an overstatement based on the results presented here. Meiotic recombination landscapes have not been analyzed/compared.

As for the section describing the LINC complex (lines 397-412) I am concerned the analysis presented are based on unpublished results. It is not clear why authors do not include the data in more detail.

Specific points:

-Lines 60-62: This statement begs for a reference.

-Lines 65-68: There is already published data referring to this.

-Lines 80-81: Please, be aware that CO distribution is sex- and species-dependant.

-Lines 557-574: The workflow described for the FACS methodology it is not entirely clear. Which animals are represented in the FACS profile depicted in figure 1b? Adults? Is so, please provide an example profile for the synchronization where leptotene stage have been isolated. Are FACS experiments always consistent? What about cell viability? Were all cells collected in a single experiment? How many animals were used in total?

-I have concerns about the purification protocol and the ability of obtaining really distinct different population during prophase I. For example, which criteria were followed to distinguish between FACS-enriched pre-L, L, Z, P and D stages? How can the authors distinguish between pre-L and L? Any specific marker? In some of the batches less than 20 cells were analyzed under the microscope (Table S1). These concerns are also raised after analyzing supplementary Figure 2, which is kind of confusing. It is not clear difference between panels 'a' and 'b'. Legends seem to have the same colour code. Also, it seems there is higher correlation between different prophase sub-stages than between sertoli and spermatogonia replicates.

-HiC libraries: It is surprising the relatively low percentage of unique valid pairs, high variable among cell types. It varies from 45% in one batch of P to 74% in D (Table S2). There are also high differences

between batches in terms of total reads in some cell types (i.e., 19M reads vs 138M reads in the case of sertoli). Can authors comment on that? Can this affect the results obtained?

-Lines 466-468. Please include more recent relevant literature: PMID: 30879787, PMID: 25590558.

-Lines 485-491. Such conclusions are an overstatement not directly obtained by the data analyzed. The same applies to the sentence 'Such an extensive correlation suggests that the spatial proximity between chromosome ends may serve as a contributing mechanism for the global modulation of CO occurrence' (lines 499-501).

Response to Reviewer for NCOMMS-20-36524A

Reviewer #1 (Remarks to the Author):

Overview: Here the authors present a detailed analysis of mouse meiotic chromosome conformation as measured by HiC. Key analytical findings are that compartment A and B domains may have differing compaction rates, perhaps suggesting differential loop size, and that this varies during the meiotic stages. The authors also demonstrate significant interchromosomal interaction of subtelomeric regions (most likely detection of a bouquet-like stage) that is largely abrogated by Sun1 mutation. The data appear high quality, are well presented, and the manuscript is clearly written. However, in a number of places, data are over-interpreted, and measurable statistical differences need to not be confused with something having a meaningful biological effect/difference.

Notably, a number of relatively high-profile meiotic HiC studies (in various organisms) were published in the past year or so in: NSMB, Molecular Cell, Cell Reports, and Nature Communications. The authors do adequately reference these studies, although I have some concerns that the new datasets and analyses that are presented in this manuscript are relatively incremental in their impact compared to those prior works. Nevertheless, overall, this is a well-constructed study, that requires, mostly, only text changes to address the concerns and questions that I have raised.

We are grateful to Reviewer #1 for the enthusiastic interest in our results and helpful suggestions. In our revised manuscript, we have addressed all the reviewer's questions by the addition and modification of requested figures, and textual changes.

General. To maximise accessibility, I would suggest that the authors do not use the abbreviations P, Z, L, preL, etc for the various stages. These seem unnecessary, frequently making the reading of the text challenging to follow.

We have now changed the abbreviations preL, L, Z, P, etc. throughout the manuscript to preleptotene, leptotene, zygotene, pachytene stages, etc. to improve the clarity.

General: For all HiC plots, please indicate in the image or legend the bin size that is being plotted/analysed.

We have now indicated the bin size for Hi-C heatmaps in figure legends throughout the manuscript.

Specific comments and suggestions:

Line 90-95. Whilst it is true that many of the prior studies included pachytene-stage samples, Patel 2019 included zygonema stage and Vara 2019 included a

Leptonema/Zygonema sample. Please revise text to ensure that prior literature is represented accurately.

We have now added the Patel 2019 and Vara 2019 papers as references in Line 93.

Line 107. Similarly, it is an unnecessary overstatement to say that these data are "highly demanded" when prior work already exists. The authors, here, present nice new data and observations that build on prior findings. There is no need to overstate things. Please recognise the prior work more thoughtfully.

We have made textual changes by removing the terms "highly demanded" and changed the statement to "Hi-C characterization in mammalian meiocytes at a greater temporal resolution, in a fashion similar to yeast, would improve the understanding of the meiotic chromosome reorganization processes in mammals"(Line 107 to 110 in the revised manuscript).

Section 130-142. A much more detailed description of the criteria used to isolate and validate the specific stage-resolved populations is required. This is a critical part of the study, yet almost no detail is presented. Specifically, what aspects of the FACS data and microscopy enable the designation of specific staged populations?

We appreciate the reviewer's suggestions. Being able to isolate the meiocytes at precise stages including preleptotene, leptotene, and zygotene is indeed one of the strengths of our study. And we have now included a much more detailed description of the cell isolation and quality assessment procedures in both the results (Lines 135 to 144) as well as the materials and methods (Lines 666 to 753).

We would like to emphasize here that we employed a brute-force approach to obtain a sufficient number of meiocytes of high purity. For instance, to isolate the preleptotene stage spermatocytes, we combined the testes of 5-8 mice at the appropriate age and used a highly stringent window for FACS sorting, which resulted in higher purity of cells but lower yield. We performed 16 independent FACS experiments for the preleptotene stage cells. After quality inspection, we kept 10 out of 16 samples that exhibited >80% viability and >78% purity and further pooled the cells into two batches for downstream studies. The 6 samples that did not meet our quality criteria were discarded. We have now compiled a supplemental table (revised Supplementary Table 1) that summarizes the information for each experiment we performed in this study, regardless of whether the batch was used for Hi-C library construction.

To further validate our experimental procedures, we repeated the spermatocytes isolation procedures and obtained two new batches of preleptotene, leptotene, and zygotene stage cells during the revision and performed bulk RNA-Seq on these cells. The transcriptome profiling showed that these isolated cells express marker genes for their respective meiotic stage that have been revealed in a previous single-cell

transcriptome study (revised Supplementary Figure 2). Taken together, we have high confidence in the purity of the stage-resolved meocytes used in our study as well as the quality of our data.

Line 148. Where does Fig 1c demonstrate telomere clustering at Zygotene? For this to be informative, a zoom-in and/or aggregation, and comparison with other stages is required. In general, it is very unclear what Fig 1c is meant to demonstrate. In particular, the text in the legend describing Fig 1c: "...shows dynamic reorganization of 3D genome architecture during meiosis" is not supported by the content of this figure. Please clarify and/or revise.

In the original Fig 1c, we attempted to demonstrate that the overall genome organization does look different between the spermatocytes of different stages using a similar layout as in the Vara et al. 2019 study. However, we realize that it may be hard to appreciate the differences between different stages without zooming into individual chromosomes. We have now modified the panels in the revised Fig 1f to show the intra-chromosome interaction heatmaps of Chr1, as well as the inter-chromosomal heatmaps between Chr1 and Chr2, both at 1Mb resolution, underneath the all by all Hi-C heatmaps. We have also revised the statements in the text and the figure legend that may lead to confusion. In particular, the "telomere clustering" in the previous manuscript referred to the X-shaped inter-chromosomal interaction pattern that became visible from the leptotema stage and on. We have now changed the phrase to "interactions between telomeres of different chromosomes".

Line 149. The correlation coefficients are almost universally high irrespective of whether the data being compared are replicates or non-replicates (this is especially true of the Pre-Leptotene through to Diplotene samples (S1b) where all values are >0.95). As such, I find it hard to be persuaded that this analysis demonstrates that the replicates are highly reproducible. For example, if (all) the samples just so happened to be full of homogenous noise (once binned at 50 kb), the data would also be correlated...but this would not in any way indicate that the samples are high quality. Thus, perhaps there is a better way to demonstrate sample similarity than such a crude global correlation? For example, perhaps there are stage-specific visible patterns that can be discerned in replicates but not in non-replicate samples.

We agree with the reviewer that the universally high Pearson correlation coefficients calculated using contact probability in each bin may not be the best way to demonstrate data reproducibility. In the revised manuscript, we have comprehensively addressed the issue by performing several analyses. First, instead of calculating the correlation coefficients using contact probability for 50 kb genomic bins, we have now increased the bin size to 500 kb and recalculated the correlation using the HiCRep packages (revised Supplementary Figure 3). By increasing bin size and hence the number of valid interactions contained in each bin, the noise levels were reduced. As a result, it becomes more clear that the datasets from the same stage tend to exhibit

higher correlation with each other than with datasets from different stages. Second, we used the correlation coefficients between insulation profiles as an additional metric to confirm the reproducibility of Hi-C replicates (revised Supplementary Figure 4a). A similar approach has previously been employed in a study from Bing Ren's lab (Gorkin *et al.* 2019 Genome Biology, PMID: 31779666). We have also included the insulation profiles for a representative genomic region in all Hi-C datasets in the revised Figure S4b. These analyses collectively demonstrate that the Hi-C datasets from the same stage are reproducible, indicating good data quality.

Line 150. Please elaborate on how Fig S2b demonstrates that the cell stages are distinct from one another? Is the difference between correlation coefficients of 0.98 and 0.96 really meaningful? It suggests that there is very little difference in HiC contact maps between stages, and/or that the staged samples are not well isolated from one another temporally. Which is it?

The high correlation coefficients between Hi-C heatmaps indicate that the spermatocytes of different meiotic stages (leptotene, zygotene, pachytene, diplotene) exhibit an overall similar chromosome organization, which is unsurprising given that the meiotic chromosomes have adopted a conformation of linearly arranged large-scale loop arrays. However, with the newly added analyses in the revised Fig S3 and S4, it can be more easily appreciated that different meiotic stages do exhibit differences in large-scale organization features, such as TAD organization as indicated by the insulation profiles.

Line 152-153. Without more convincing evidence, this comes across as an overstatement.

We agree with the reviewer and we have now removed this statement.

Fig 2a-d and associated text (Lines 159-190). The curves are very similar and overlap in complex patterns. As such, please present separate curves for each replicate and/or some indication of the variance between the repeats. Adding this information will enable reader to appraise how much variation there may exist between biological replicate samples of the same stage relative to different biological samples.

In the revised manuscript, we included new figure panels in revised Supplementary Figure 5 to show the P(s) curves for the two replicates for all eight stages. Our data show that the biological replicates of the same stage exhibit highly similar scaling curves, further attesting to the quality of our Hi-C datasets.

If we assume that there is a difference in compaction at different stages as measured by P(S) in panels 2b-d (but not on Chromosome X), it would be useful to determine and present whether this compaction difference arises in all autosomes, or only in some.

We have now performed the P(s) and the P(s) slope analyses on each autosome separately (revised Fig S6-9). Our data suggest that all individual autosomes exhibit a progressive increase in average loop size in a fashion similar to the combined autosomes (Fig 2 a-d), while the X behave slightly differently (revised Fig S10).

Lien 189. Please clarify in the text why Fig S1a is referred to here.

The silencing of the X chromosome is associated with the accumulation of γ -H2AX on the X and Y chromosomes during the late meiotic prophase, which can be seen on the microscopic images in Fig S1a. However, we realize that referring to those images may not be necessary for the statement in the text and cause confusion. We have now removed the figure reference.

Line 189-190. Without additional experiments and observations, there is nothing yet presented in the manuscript that "reveal[s] that progressive loop extrusion occurs throughout the entire meiotic prophase I". The data *may* be consistent with this interpretation/model, but the data far from "reveal" it to be the case. There could easily be other mechanisms that alter chromosome conformation. In particular, there is not, to my knowledge, direct evidence yet of loop extrusion in meiotic cells.

We agree with the reviewer that we have only shown that the average size of meiotic chromosome loops gradually increases, but did not unveil how this phenomenon occurs. Indeed, there has been no direct experimental evidence showing loop extrusion during meiosis, particularly in mammals. That being said, a recent polymer simulation study (Schalbetter et al. 2019) suggested that the meiotic Hi-C patterns in yeast agreed with the loop extrusion model. Besides, given that the condensin-driven loop extrusion has been shown to orchestrate the organization of the mitotic chromosome, it is plausible that a similar loop extrusion mechanism may control the assembly of meiotic chromosome loops. We have now revised the statement to "which may result from the loop extrusion process driven by the meiotic cohesin complex" (Lines 217 to 218). Also, since the term "loop extrusion" has been tightly associated with the SMC-mediated loop extrusion model nowadays, to avoid unnecessary confusion, we have removed or modified the usage of "loop extrusion" at a number of places throughout the results.

In Fig S4a, please could the positions of the boundaries be more clearly indicated on the axes? The relevant tick marks are currently almost invisible. Moreover, it would be helpful if these plots were represented as fold changes compared to expectation (essentially a genome average contact matrix), since this will normalise for the expected interaction all-off that happens with increasing distance from the diagonal, and thus should emphasise any boundary features.

We appreciated the reviewer's suggestion. We have now remade the TAD pile-up

heatmaps in the revised Supplementary Figure 11h. For this new set of figures, we generated the pile-ups using the 1Mb genomic regions centered at TAD boundaries and normalized the observed Hi-C interactions using the expected interaction probabilities at each genomic distance.

Line 201-203. And Fig S4c. How many of the boundaries overlapped in each stage? It is possible to present a Venn diagram of these data? i.e. Are the boundaries that are present in the L–D samples mostly the same? And are they a subset of the boundaries detected in the Sertoli etc cells, or completely different?

We appreciate the reviewer's suggestion and have now supplemented this information in the revised Supplementary Figure 11a-f. In short, we found the locations of TAD boundaries are variable between stages. Using a 5-way Venn diagram, we show that the boundaries in meioocytes (leptotene to diplotene) overlap to a higher extent with each other than with the Sertoli cells (Supplementary Figure 11a). We also generated Venn diagrams that illustrate the overlapped TAD boundaries between successive meiotic stages (Supplementary Figure 11b-f). We found that about 2/3 of TAD boundaries are shared between successive stages. However, we would like to point out that the insulation profiles for the two replicates of the same stage exhibit similar patterns (Supplementary Figure 4b), suggesting these variations are likely to be biological. The underlying causes for these variations are unclear.

Line 206. It is not clear that the boundaries have "marked weakening" in PreL stage. Fig S4c seems to indicate weakening at the spermatogonia stage. Please clarify and/or revise.

In response to the reviewer, we have remade the TAD boundary pile-up heatmaps in revised FigS11h and quantified the strength of TAD boundaries using boxplots in revised FigS11g. Both figures indicate the weakening of TAD boundaries at the spermatogonia stage as well as in the preleptotene stage. We have also revised the text from Line 232 to Line 238 to describe the changes in TAD boundaries more accurately.

Line 216-220. Please revise main text to make it clear that this analysis was performed using the pachytene stage data.

We have added the sentence "Previously published ChIP-Seq data revealed a large number of REC8 peaks in mouse pachytene/diplotene spermatocytes " to indicate the stage of REC8 ChIP-Seq data. (Lines 256 to 257)

Line 219-220. Given that there are approximately three times as many REC8-only sites as REC8+CTCF sites, is it reasonable to conclude that, even despite lower peak ChIP signal, that the REC8-only sites are the "preferential binding sites"? i.e. It seems entirely possible that, globally, there is more REC8 at non-CTCF sites than at CTCF

sites... Moreover, REC8 peaks (whether at CTCF or not) may represent only a tiny fraction of all the, potentially heterogeneously-bound, REC8.

We agree with the reviewer that the total amount of REC8 bound to these CTCF sites may only account for a fraction of the total REC8 binding events. And we have no intention to make an impression that the meiotic chromatin loops are predominantly attached to these sites. However, the occupancy of REC8 is significantly higher at these sites than any other set of genomic locations, indicating higher probabilities for these sites to be associated with the chromosome axis. To avoid confusion, we removed the terms “preferential binding sites” and added the following sentence “Although the binding of REC8 in mammals is likely to be stochastic among cells, these CTCF/ REC8 co-occupied peaks may represent locations where the chromatin is associated with the axis at a higher frequency than the rest of genome.” (Lines 263 to 266)

Line 221-222. Please indicate the total number of CTCF peaks analysed in the two classes. Indeed, for clarity, perhaps the number of peaks can be added to both Fig 2e and 2f.

We have added a Venn diagram of REC8 and CTCF peaks (revised Fig 2f) to indicate the number of peaks in each category.

Fig 2g-h, For clarity, please label the cell stages that were analysed in the title of these figures (as in 2e-f), rather than in the Y-axis.

We have now generated a new set of figures illustrating the changes in insulation ability at these peaks and moved them to the supplemental figure (revised FigS13). The cell stages are now labeled in the title of these figures.

Line 231-234. Please revise this text, since the conclusion does not seem to make sense. If the insulation score goes up more, how/why does it indicate greater *loss* of insulation ability?

We realized that our mixed use of “the insulation score” and the “boundary strength” in the text may have caused confusion. The insulation score reflects the aggregate of interactions occurring across each genomic interval. Minima of the insulation profile, which are usually negative values, denote areas of high insulation which represent TAD boundaries. Therefore, the increase of an insulation score indicates the loss of insulation ability at a genomic interval, hence the weakening of a TAD boundary. In contrast, the “TAD boundary strength” scores that were shown in the original Figure S4c (now Figure S11g) quantify the insulation score difference between a TAD boundary and its surrounding regions. The “TAD boundary strength” scores are positive values, with the greater values indicating “deeper valleys” on the insulation profile hence the stronger TAD boundaries. While the “boundary strength” provides a

more intuitive indicator for the boundary strength, it is only calculated at the genomic intervals identified as TAD boundaries from the cooltools package. On the other hand, the “insulation scores” can be calculated at every genomic interval. We have now revised the text and figure legend to clarify the meaning of these different metrics.

Secondly, whilst the difference between the red and orange distributions is significant, this is likely to be only because there is so much data. i.e. My point is this: Quantitatively, is there really any substantial difference?

We thank the reviewer for raising this point. Indeed, we need to be cautious about the small p-values associated with a large sample size and avoid overinterpreting these results. In our revised manuscript, we have chosen not to emphasize these points and moved them to Fig S13, where we described the patterns of the insulation changes at different sites in detail in the figure legend. That being said, since genomic intervals with insulation scores less than -0.1 were usually reliably identified as TAD boundaries, we would like to think that the changes of insulation scores at a comparable magnitude are likely to be biologically relevant. For instance, the CTCF/REC8 peaks exhibit a median insulation increase of 0.36 at the preleptotene stage compared to in somatic cells, while the CTCF-only peaks exhibit a median insulation increase of 0.30 (Fig S13c). Such a difference could reflect different chromatin structural changes associated with these sites upon meiosis entry.

Line 252-253. Please can the authors clarify in the text what they mean by "more dynamic"? Do the authors infer greater heterogeneity in loop positions to mean greater dynamism? Is there evidence to support this? Loops in interphase could be well-positioned across a population, yet highly dynamic, for example.

We thank the reviewer for pointing out this issue. We indeed wanted to make a point that the meiotic chromosome loops are heterogeneously positioned and are not anchored at a specific set of locations. However, our choice of word may have caused unnecessary confusion, as ‘dynamic’ infers that these loops are actively forming and deforming over time. We have modified the statement to “Nonetheless, our data suggest the high occupancy of CTCF and REC8 at these sites do not create a strong barrier for meiotic loop extension. The inability to immobilize chromatin loops at specified locations may contribute to heterogeneity in meiotic chromatin loop sizes during the progression of meiosis.” (Lines 303 to 307).

Fig S5a. Perhaps the y-axis should be relabelled as percentage of maximum (as observed in PreL)? At present it is rather odd to have $A+B=200\%$. What are the absolute proportions of compartment A and B (i.e. is A or B more frequent?) This information appears to be hidden for no obvious reason.

We appreciate the reviewer for this suggestion. We have remade the bar plots to show the changes of the A and B compartment more clearly (revised Figure S15a, b). In Fig

S15a, we compared the compartment composition in each stage with the preleptotene stage, in which 46% of the genome belongs to the A compartment and 54% of the genome belongs to the B compartment. We have also assessed the changes of the A/ B compartment between successive meiotic stages (revised Figure S15b).

Lines 270-272. Fig S5a. It appears that it is specifically B that reduces during Zygotene to Diplotene transition. Perhaps this should be commented upon?

Indeed, when compared with preleptotene, a higher fraction of genome switched from B compartment to A compartment in pachytene and diplotene stages (11% for pachytene and 12% for diplotene) than genome switched from A to B (9% for both pachytene and diplotene), resulting in an increase of B compartment. We have commented on this trend by adding the following sentence “Interestingly, after the meiosis entry, the fraction of genome belonging to active A compartment first decreases in leptotene and zygotene stages, and subsequently increases in pachytene and diplotene stages.” (Lines 325 to 327)

Line 270-271. Please elaborate in the text on how Fig 3a demonstrates gradual attenuation of compartmentalization from PreL to D. PreL looks clearly different (but please explain in what manner). The other stages look more or less indistinguishable in this representation.

From Fig 3a, we think it can be appreciated that the checkerboard pattern was attenuated in leptotene and diplotene compared to preleptotene, and further diminished in pachytene and diplotene. However, we agree with the reviewer that the differences between leptotene and diplotene, and between pachytene and diplotene were not evident. We have changed the terms “gradual attenuation” and revised the statement to “Upon meiosis entry, the extent of genome compartmentalization, which is indicated by the checkerboard pattern on Hi-C heatmaps, exhibited a marked decrease in leptotene and zygotene stages, and further diminished in pachytene and diplotene (Fig. 3a).” (Lines 329 to 332)

Line 271-272. Please elaborate in the text how Fig S5b supports the statement that: "regions of different transcriptional status rarely intermingle with each other".

Fig S5b (now Fig S15c) depicts the interactions between two genomic regions separated by 40 Mb (Chr6: 25M- 31M and Chr6: 79M- 85M), thus representing an off-diagonal region from the Hi-C heatmap. The purpose of this figure is to show that the long-range association of regions belonging to A compartment or belonging to B compartment diminished during meiosis. Meanwhile, the long-range association between regions belonging to different compartments did not increase, suggesting that the weakening of compartmentalization was not accompanied by increased inter-compartment intermingling. We have now revised the text to clarify these points by adding the following sentences “Such an attenuation in compartments can be

partially attributed to the condensation of chromosome loop arrays and the global loss of long-range chromatin interactions. While the long-range association between genomic regions belonging to the same compartment decreased, the interactions between regions belonging to different compartments remained at a low level (revised Fig S15b,c). These findings suggest that the transcriptionally active and inactive regions form alternating, large-scale chromatin domains along the meiotic chromosome axes, consistent with the previous cytological observations.” (Lines 336 to 339)

Line 273-274. Where is the evidence for this statement? The boundaries between A and B are not discernible in the raw data in Fig S5b L to D stages.

We would like to clarify that Fig S5b (now Fig S15c) shows the off-diagonal interactions between two far-apart genomic regions and thus does not indicate the boundaries between the A and B compartment. We made the statement based on the eigen1 tracks (revised Fig S14) that exhibit alternating A and B compartment patterns. In addition, the domain boundary in Fig 3b also persists through meiosis. Such a finding is also consistent with the previous EM observations that the G/R band form separate domains on meiotic chromosomes. We have now modified the statement to “These findings suggest that the transcriptionally active and inactive regions form alternating, large-scale chromatin domains along the meiotic chromosome axes, consistent with the previous cytological observations”.

Lines 276 to 282 and Fig 3b-f. This is a very nice set of analyses. Very intriguing!

We appreciate the reviewer's enthusiasm for these analyses! We do hope that these findings could offer new insights into the field.

Line 313-314. Could the authors elaborate in the text why they think more DNA near the chromosome axis would aid homolog recognition? This isn't obvious to me.

Since the DNA recombination machinery preferentially resides on the chromosome axis, it is conceivable that the higher fraction of DNA located near the axis would result in a higher fraction of DNA having access to the recombination machinery, thereby resulting in higher chances of forming inter-homolog links. We have now revised the text to explain these points more clearly. (Lines 377 to 382)

Line 328 and Figure 4c. As mentioned above, whilst highly statistically significant (due to lots of data), the actual absolute difference here in "alignment precision score" between A and B is extremely small. I would caution the authors not to over-interpret the significance of such tiny differences. i.e. the important point here is not the statistical significance, but the effect size (which is actually tiny).

I have a similar concern about Fig 4e. These data distributions, whilst apparently

statistically different, are also incredibly similar. So does this really mean anything?

We think the reviewer's concerns regarding Figure 4c and 4e are valid. Indeed, the differences in alignment precision scores are very small despite significant p values. To avoid overinterpretation of these data, we have now removed the "alignment precision score" from our analysis. However, we have included a new set of analyses that reveal that the variations in chromosome loop sizes among genomic regions correlate with the rate of crossover formation.

Line 361. and Line 362-364. Where in Fig S1a is the data to back up this statement that <5% of the cells in Zygotene exhibited bouquet conformation? Importantly, consider how were the IF images prepared. Methods indicate that chromosomes were spread. Does this maintain or does this disturb bouquet structure? I would suspect the latter... Without further evidence one way or the other, the conclusion(s) in this section need substantial revision.

We agree with the reviewer that the chromosome spread procedure may disturb the bouquet structure. During the revision of our manuscript, we performed the immunofluorescence on the FACS-isolated zygotene spermatocytes without treating the cells with hypotonic buffer and performing the chromosome spread procedure. The conformation of chromosome ends is expected to be preserved better in these non-spread cells. We imaged ~180 zygotene spermatocytes and only found 6 cells exhibiting large clusters of chromosome ends, which may indicate the bouquet conformation. However, we could observe small clusters of chromosome ends that usually consisting of 5-6 clusters in a larger fraction of cells. Thus, we conclude that the zygotene cells isolated in this study do not enrich for cells exhibiting bouquet arrangements. Therefore, the alignment of chromosome ends described in our paper is unlikely to be strictly caused by the formation of the bouquet.

Line 397-412. It is rather hard to appraise this section since so many of the observations that support the ideas are listed as "unpublished". If they are relevant to the logical flow of this manuscript should they not be included here?

In a separate study, we discovered that the connection between telomeres and the LINC complex was mediated by the interaction between SUN1 and Speedy A (SPDYA). The disruption of SUN1-SPDYA interaction by the SUN1 W151R mutation led to severe meiotic defects. By performing Hi-C on the SUN1 W151R mutant spermatocytes and comparing it with the published data from Patel *et al.*, we showed the mutation caused decreases in the interaction between telomeres. Since a manuscript describing those results (referred to as "Chen *et al.* unpublished" previously) was already under review when we initially submitted our manuscript, we felt it would be inappropriate to include similar data in our manuscript. The Chen *et al.* study has recently been accepted by Nature Communications. We have now attached the accepted manuscript by Chen *et al.* (NCOMMS-20-26947A, "The SUN1-SPDYA

interaction plays an essential role in meiosis prophase I”) together with our revised manuscript for your evaluation.

Line 423-430. It should perhaps be mentioned that at the very tips of the chromosomes the observed interchromosomal interaction frequency is now *increased* in the Sun1 mutant. Thus contacts are both increased, yet persist of a shorter fraction of the chromosome length.

We appreciate the reviewer’s suggestion, we have now added the following sentences “Notably, the association among the very tips of chromosomes even increased in Sun1 mutant compared to the wild-type spermatocytes” And “..., but the contacts at the chromosome ends persist to a shorter fraction of chromosome length” to the results (Lines 513 to 522).

Line 448. Please revise the mention of dynamic loop extrusion in line with comments made above.

We have re-written the paragraph in the discussion related to the meiotic loop extrusion (Lines 540 to 557).

Reviewer #2 (Remarks to the Author):

In this work, Zuo and collaborators mapped the 3D genome architecture through spermatogenesis. Obtaining HiC maps from highly differentiated cells such as germ cells is quite a challenge so I acknowledge the effort to conduct the work presented here. Also, the paper is well written given the complexity of the data generated.

We appreciate Reviewer #2's positive evaluation of our work and helpful suggestions. In our revised manuscript, we have addressed all the reviewer's concerns by the addition of new tables, figures, and changes to the text. In addition, we have also performed new experiments and analyses such as profiling the occupancy of CTCF through different meiotic stages using CUT&Tag, and examining the relationship between variations in chromatin loop sizes and the occurrence of CO events, which we think has improved the quality and depth of our manuscript.

Given said that, my major concern relay on the novelty of the data presented. The 3D maps provided in the present manuscript are in fact patterns have already been reported in prior studies (Wang et al. 2019; Patel et al. 2019; Alavattam et al. 2019; Vara et al. 2019). It is true that pre-leptotene stage and Sertoli cells maps are new in the literature, but it is not clear how authors are able to differentiate among all cell types during prophase I. In fact, I have concerns regarding the purification process (see specific comments below).

Also, the description of the progressive increase of chromosome loop size throughout meiotic prophase I (lines 158-190) have also been reported previously, not only in mammals but also in yeast. The same occurs for interactions at telomeric regions (lines 337-348).

We acknowledge that in the recent two years, while we were conducting our study, several studies have profiled the chromosome organization of spermatocytes at various meiosis stages: Wang et al. 2019- Pachytene stage; Patel et al. 2019- Zygotene and Pachytene stages; Alavattam et al. 2019- Pachytene stage; Vara et al. 2019- mixed Leptotene/Zygotene stages and mixed Pachytene/Diplotene stages. All of these studies revealed the most obvious changes in meiotic chromosome organization, namely the loss of TAD organization and the attenuation of the A/B compartment, and discussed their functional implications in transcriptional regulation. Compared to these previous studies, our Hi-C dataset that consists of Preleptotene, Leptotene, Zygotene, Pachytene, and Diplotene stages offers a full picture of dynamic changes of meiotic chromosome organization. Our study, built on the previously published findings, provides new observations that further link the meiotic chromosome structure with meiotic cellular functions, particularly the homologous recombination. While the concepts of progressive increase of chromosome loop size and the coalescence of telomeres are not entirely new, we presented a more thorough description of these two

phenomena than all the aforementioned studies in the result sections (Figure 2a-d and Figure 5a-d). Describing these results set up the contexts for the two major new observations made in this study, which we elaborate below:

One of the key findings from our study is that the genomic regions exhibit variable chromatin loop sizes during leptotene and zygotene stages, when homologous recognition and recombination happens, and that the variations in chromatin loop size are correlated with the transcriptional activities and A/B compartment identity. These variations have not been quantitatively analyzed previously. We would like to emphasize that the loop size differences only exist in early meiotic prophase I, when homologous recombination occurs, but diminished in pachytene and diplotene stages, thus highlighting the values of our stage-resolved datasets. Importantly, by performing a new set of analyses, we reveal a link between the average loop sizes in different genomic regions and the occurrence of crossovers. These findings collectively suggest a model in that the variability in transcription activities, through the modulation of large-scale chromatin organization, could influence the distribution of meiotic recombination events.

Our study also provides new insights for understanding the spatial organization of chromosome ends during meiosis. While the association between telomeres has been demonstrated in numerous studies, not only by Hi-C analyses but also by cytological and genetic studies, we focused on the interactions between different chromosome ends that persist over ~20% of the chromosome length, which has not been explicitly studied before. Such a window of chromosome ends “alignment” correlates with previously reported windows that exhibit higher crossover occurrence during spermatogenesis. We further showed that the extended alignment of chromosome ends, but not the clustering of telomeres at the very tips of chromosomes, were abrogated by a mutation in the LINC complex. These findings suggest that the LINC complex exerts a unique function during meiosis to promote the alignment of subtelomeric regions over a substantial range, and provide a potential link between LINC-mediated force transmission and the modulation of meiotic recombination.

With regard to our purification procedure, we appreciate Reviewer #2 for bringing up the issue. Since being able to isolate different testicular cell types, particularly the cells representing different meiotic prophase I substages, is a critical part of our study, we felt that we did not do a good enough job to describe our purification workflow in the previous manuscript. We have now included a more detailed description of our experimental procedure in the methods (lines 666 to 753 in the revised manuscript), and added more information to the supplementary table 1. Importantly, we have also assessed the purity of the isolated cells by performing bulk RNA-Seq on two independent replicates of preleptotene, leptotene, and zygotene stage spermatocytes, which are the most challenging meiotic stages to separate from each other. Our purified spermatocytes exhibited characteristic transcriptional signatures that have been revealed previously by single-cell RNA-Seq, further

demonstrating the efficacy of our cell-isolation strategies (Revised Supplementary Figure 2). We will further elaborate on the issues related to cell isolation in the responses to Reviewer #2's specific questions below.

Later on the manuscript (lines 192-255), authors claim that anchoring chromatin loops by REC8 enables dynamic regulation of loop extrusion during meiosis, but based on the results presented it is not clear how the authors arrive to that conclusion. Also, authors talk about TADs and loops, but are they able to distinguish between these two genomic features based on their data?

We thank the reviewer for raising this point. In fact, Reviewer #1 had a similar concern. We think our use of the phrase “dynamic regulation” was unclear and has caused confusion. The point we wanted to make was that anchoring chromatin loops by REC8 does not immobilize the loop during the progression of meiosis, as opposed to the loop extrusion barrier function of CTCF during interphase. This leads to the heterogeneous positioning of meiotic loops as opposed to the well-positioned interphase loops. The logic that leads to this statement is as the following: The meiotic chromatin loops progressively extend to sizes of 1.4 to 1.6 Mb, which exceed the separation between the > 6000 CTCF/REC8 co-occupied sites. If these axis-tethering sites possess the ability to immobilize loops hence halting the extension of loops, the population-averaged interaction frequencies among these sites should eventually increase at late meiosis. However, we did not observe the increase of interactions between these CTCF/REC8 co-occupied sites in pachytene and diplotene, suggesting that these sites do not confer strong constraints on chromatin loops and may be bypassed during the loop extension.

Regarding the use of terms of “TADs” and “loops”, we do realize that the general term “TADs” only stands for any self-interacting domains that appear in Hi-C heatmaps as squares around the diagonal and that TADs can be generated via several different mechanisms. However, TADs that are demarcated by CTCF binding sites usually correspond to the looping events between these sites, which occur at a higher frequency due to the extrusion barrier function of CTCF. Therefore, we also use the phrases “CTCF-anchored loops” when describing the interactions between CTCF-bound sites.

In addition, the authors conclude that transcription-coupled variations in chromatin loop sizes during early meiotic prophase (lines 257-306). But it is not clear which transcription data are the authors considering for their analysis. While A/B compartments are defined based on eigenvector (what the authors are studying here) active transcription regions are based on RNA-seq data. Often both features are correlated but this needs to be proven.

We agree with the reviewer that the correlation between the A/B compartment and transcriptional activity needs to be demonstrated. During the revision, we have

generated bulk RNA-Seq data for preleptotene, leptotene, and zygotene stage spermatocytes. Supplementary Figure 14 in the revised manuscript indeed indicates an overall correlation between the RNA-Seq data and the Eigen1 values.

Authors also state that ‘... our analyses reveal an extensive correlation between transcription-coupled variations in chromatin loop size and the extent and precision of homolog alignment, implying that chromatin loop sizes may contribute to the modulation of meiotic recombination landscape’ (lines 32-335). I think this is an overstatement based on the results presented here. Meiotic recombination landscapes have not been analyzed/compared.

In the revised manuscript, we have performed new analyses to examine the relationship between meiotic chromatin loop sizes and the distribution of recombination events (revised Fig 4d-h). By calculating the DSB and CO densities in each A/ B compartment region using previously published data, we found that the A compartment regions that enrich in shorter-range chromatin interactions exhibit a higher crossover/ DSB ratio than the other regions. These results imply that the structural properties of loop arrays could impact the probabilities of converting DSBs into COs.

As for the section describing the LINC complex (lines 397-412) I am concerned the analysis presented are based on unpublished results. It is not clear why authors do not include the data in more detail.

In the LINC complex section, we employed a mouse model in which the interaction between telomere and the LINC complex was disrupted by a single point mutation in SUN1 (SUN1 W151R). The manuscript describing the LINC mutant mouse model (referred as “Chen *et al.* unpublished” previously) was under review at Nature Cell Biology when we first submitted our manuscript, was later transferred to Nature Communications and has now been accepted by Nature Communications. We did not include the data describing the meiotic defects caused by the SUN1 W151R mutation to avoid data duplication in our initial submission. We have now attached the accepted manuscript by Chen *et al.* (NCOMMS-20-26947A, “The SUN1-SPDYA interaction plays an essential role in meiosis prophase I”) together with our revised manuscript for your evaluation.

Specific points:

-Lines 60-62: This statement begs for a reference.

We refer to the Prakash *et al.* 2015 PNAS paper here (PMID: 26561583), which showed that the histone mark H3K4me3 associated with active transcription emanates radially from the axis of the SC during pachytene using superresolution microscopy.

-Lines 65-68: There is already published data referring to this.

We thank the reviewer for pointing this out. We have now revised these sentences to: “The sex-dependent differences in chromatin loop sizes and axis lengths correlate with the sexually dimorphic CO frequency, suggesting that the meiotic chromosome organization may contribute to the global modulation of recombination events. However, whether and how the local variations of meiotic loop array organization across the genome influence homolog interactions and recombination frequency have not been fully understood.” (Lines 65 to 70 in the revised manuscript).

-Lines 80-81: Please, be aware that CO distribution is sex- and species-dependant.

We have now revised the text to “it has been documented that COs preferentially occur near telomere during spermatogenesis” (Lines 82 to 83 in the revised manuscript).

-Lines 557-574: The workflow described for the FACS methodology it is not entirely clear. Which animals are represented in the FACS profile depicted in figure 1b? Adults? Is so, please provide an example profile for the synchronization where leptotene stage have been isolated. Are FACS experiments always consistent? What about cell viability? Were all cells collected in a single experiment? How many animals were used in total?

We do realize that we did not describe our cell isolation and quality assessment procedures clearly enough in our original manuscript. We would like to clarify that we actually used animals of different ages for isolating spermatocytes at different stages: 7-11 days old mice were used for isolating Sertoli, spermatogonia, and preleptotene cells, 16-day-old synchronized mice for leptotene cells, 2-week old mice for zygotene cells, 3- to 5-week-old mice for pachytene, diplotene, and Metaphase II cells. The original Figure 1b showed a representative FACS profile for testicular cells from 3-week-old mice, which contain cell populations of all meiotic stages. The FACS profile from 3-week-old mice depicts that the different cell types, such as Sertoli, spermatogonia, preleptotene, leptotene/zygotene (L/Z), pachytene/diplotene (P/D), and Metaphase II cells, occupy different regions. However, we found that using younger mice that enrich for spermatocytes at earlier meiotic stages did improve the purity of the isolated cells. We have now remade Figure1 and included representative FACS profiles for 10-day-old, 2-week-old, and 3-week-old unsynchronized mice as well as 16-day-old synchronized mice (Fig1b-e).

We did not purify all cells for a specific stage from a single experiment. In fact, since we used quite stringent gating strategies, we traded the cell yield off for higher cell purity. As a result, we had to perform many experiments to obtain ~1 million cells for each Hi-C experiment. For isolating pachytene and diplotene cells, we used one or two 3- to 5-week-old mice in each experiment. For earlier meiotic stages, we used the

testes from up to 8 younger mice in each FACS experiment. We found that the FACS profiles from different experiments for mice of the same age are usually consistent. And the viability of cells was usually higher than 80% using our purification procedure. However, to ensure maximal purity and cell viability, we performed cell quality assessment for each FACS experiment, which we have described in more detail in the revised Methods section. Briefly, after FACS sorting, we checked cell viability and count the cell numbers, and used 15-20 thousands of cells for chromosome spread. Patterns of SYCP3, γ H2AX, and DAPI were used to categorize the cells to different meiotic stages. After the quality assessment, the isolated cell populations that were of > 80% viability and >78% purity were saved and further pooled into two batches for Hi-C library construction. The isolated cells that did not meet our selection criteria were discarded. For instance, to isolate sufficient preleptotene stage spermatocytes, we performed a total of 16 FACS experiments. Cells from 10 experiments were saved and 6 experiments were discarded. The information for all the experiments is now included in Supplementary Table 1. In summary, a total of 223 C57BL/6J mice were used to isolate sufficient cells for generating the Hi-C datasets in this study. And this does not include a significant number of mice used in pilot experiments to determine the optimal gating areas for mice at different ages.

-I have concerns about the purification protocol and the ability of obtaining really distinct different population during prophase I. For example, which criteria were followed to distinguish between FACS-enriched pre-L, L, Z, P and D stages? How can the authors distinguish between pre-L and L? Any specific marker? In some of the batches less than 20 cells were analyzed under the microscope (Table S1). These concerns are also raised after analyzing supplementary Figure 2, which is kind of confusing. It is not clear difference between panels 'a' and 'b'. Legends seem to have the same colour code. Also, it seems there is higher correlation between different prophase sub-stages than between sertoli and spermatogonia replicates.

We categorized the meiotic prophase I spermatocytes using the following criteria: Cells were categorized as preleptotene spermatocytes if they exhibited weak, diffused, or punctate SYCP3 but no obvious stretches of SYCP3 signals. Cells showing short to long SYCP3 stretches that do not span over the entire length of the chromosome are characterized as leptotene cells. Cells were characterized as zygotene cells if they exhibited long SYCP3 fibers that indicate the formation of chromosome axes, as well as strong γ H2AX signals, but did not exhibit the XY sex body that is intensely labeled by γ H2AX. Pachytene cells could be identified based on the "stubby" SYCP3 signals that indicate the fully synapsed tetrad chromosomes, as well as the appearance of the γ H2AX-labelled sex body. Finally, cells were categorized as diplotene if they exhibited partially separated chromosome axes. Although we used 15-20 thousands of cells from each experiment for chromosome spread, we found the number of cells that remained on slides after the entire immunofluorescence procedure is usually low. Notably, in several pachytene and diplotene isolation experiments, there were less

than 20 cells remained on slides. However, the identity of pachytene and diplotene stage cells can usually be identified unambiguously. And those experiments with a low number of cells that remained on slides all exhibited high purity. Therefore, we decided to keep those cells for library construction.

In previous supplementary Fig 2, panel b was a subset taken from panel a that only contains the meiotic prophase I substages and was drawn using a different color scale. However, a wrong color bar was placed on FigS2b in the previous submission. We have now extensively addressed the reproducibility issues between our Hi-C replicates. First, we found that the universally high Pearson correlation coefficients between Hi-C datasets were partially caused by the use of 10kb bin sizes, which led to a large number of genomic bins containing low amounts of Hi-C contacts. This issue can be mitigated by using a larger bin size. In our revised manuscript, we assessed the correlation between Hi-C interaction matrices binned at 500 kb using the HiCRep package (revised Supplementary Figure 3). The high degrees of correlation between biological replicates, as well as the differences between the Hi-C data of different meiotic stages, can now be more readily appreciated. Second, we have also calculated the Pearson correlation coefficients between the pairwise combination of the insulation profiles, which further demonstrates the good reproducibility between our Hi-C datasets (revised Supplementary Figure 3).

While the correlations between Sertoli replicates (0.91 for interaction frequencies and 0.87 for insulation profiles) are lower than the correlations between spermatocytes replicates, they are significantly higher than the correlations between Sertoli samples with any other stage. We are confident this would not affect the downstream analyses and the conclusions in this study.

-HiC libraries: It is surprising the relatively low percentage of unique valid pairs, high variable among cell types. It varies from 45% in one batch of P to 74% in D (Table S2). There are also high differences between batches in terms of total reads in some cell types (i.e., 19M reads vs 138M reads in the case of sertoli). Can authors comment on that? Can this affect the results obtained?

As the reviewer pointed out, the percentage of unique valid pairs varies across our Hi-C libraries. Further inspection of the mapping statistics suggests that such variations mainly arise from the different degrees of dangling ends or self-circles. In the revised manuscript, we have extensively examined the data reproducibility between Hi-C datasets not only by performing the correlation analysis as described above but also by inspecting the patterns of chromosome organizational features such as TADs (revised Supplementary Figure 4b) and A/B compartments (revised Supplementary Figure 14a-c). We found the Hi-C replicates with different percentages of unique valid pairs, such as the two zygotene replicates with ~30% and ~50% unique valid pairs and the two pachytene replicates with ~50% and ~70% unique valid pairs, exhibit nearly identical patterns for all structural features. We thus

conclude that the technical variations that caused the different percentages of unique valid pairs were unlikely to affect our conclusions.

That being said, we did realize that some of our Hi-C datasets, particularly the non-meiotic cells, contain a lower number of reads than others. During the revision, we have prepared new Hi-C libraries and replaced one Sertoli, two spermatogonia, and one Metaphase II datasets with newly generated datasets. All analyses in the revised manuscript involving the aforementioned stages have been redone with the updated data, and all of our conclusions remained unchanged.

-Lines 466-468. Please include more recent relevant literature: PMID: 30879787, PMID: 25590558.

We thank the reviewer for suggesting these important references. We have now cited both references in the discussion by adding the following sentences: “Moreover, a recent study revealed that the frequencies of COs covary across different chromosomes in individual nuclei⁶⁸. Importantly, the CO covariation can be explained by the per-nucleus covariation in chromosome axis lengths, strongly suggesting the regulation of axis length and/or chromatin loop size may act as a general mechanism for the global modulation of COs⁶⁹.” (Lines 564 to 569 in the revised manuscript).

-Lines 485-491. Such conclusions are an overstatement not directly obtained by the data analyzed. The same applies to the sentence ‘Such an extensive correlation suggests that the spatial proximity between chromosome ends may serve as a contributing mechanism for the global modulation of CO occurrence’ (lines 499-501).

Given the results from our new analyses, we have modified the previous statement at Lines 485-491 to “Here we show that the genomic regions enriching for short chromatin loop sizes do not exhibit a higher density of DSB hotspots. However, these regions do exhibit higher ratios between COs and DSBs, implying that the structural properties of loop arrays could impact the probabilities of converting DSBs into COs. Together, our findings suggest a model in that the high transcriptional activity could influence chromatin loop sizes during early meiotic prophase I, creating a large-scale chromatin conformation that favors homolog recombination.” (Lines 593-600).

We have also modified the statement at lines 499-501 to “implicating the regulation of chromosome conformation and spatial proximity between chromosome ends as a contributing mechanism for the modulation of CO occurrence.” (Lines 608-610).

REVIEWER COMMENTS

Reviewer #1 (Remarks to the Author):

Comments on the revision: NCOMMS-20-36524A

Overall, the manuscript is much improved, and incorporates a wealth of data and detailed analyses. The author's should be commended for the depth of their study, and the thoughtful modifications made to the text and figures during revision. A number of small outstanding points/suggestions remain in the revision. It would be helpful if these could be addressed prior to publication.

The explanation of the cell sorting and data presentation in Figure S1 (with the addition of panel b) is much improved. The new RNA-seq data in Fig S2 also show convincing differences between the sorted cell populations. Finally, the updated correlation matrix (at 500kb) better represents the differences (and similarities) between the sorted cell populations. This part of the manuscript is now greatly improved and much more informative.

Whilst Fig 1 (and especially 1f) is greatly improved, I remain sceptical of the comment about the "x-shape" and "coalescence of telomeres of different chromosomes" as presented in this figure. The authors may believe that this should be the case, but the evidence (as presented in the figure) far from supports this, and certainly the pattern (as presented) is not "prominent". All I observe is an increase in trans interactions in the centre of the plot between Chr1 and Chr 2, not an "X". The much better analysis (and evidence) of this is presented in Figure 5, so I would suggest that the authors tone down this section (since, without evidence within the figure it is confusing to describe in this manner). The authors could optionally mention that this will be looked at in greater detail later on. It would also be helpful to indicate, on these zoom ins, which end of the chromosome is the centromere is (since these are telocentric chromosomes).

Given that each chromosome is now presented separately in Fig S6-9 (which is very helpful and convincing!), it would be far more helpful for the reader (in terms of drawing comparisons), for the X-chromosome data currently on Figure S10, to be included as relevant panels in Figs S6-9. There is even a nice empty space for the X-chromosome panels in the bottom right of each figure. Importantly, overall (and certainly as presented!) I think it is hard to make the case that chromosome X is really any different than any one of the individual autosomes.

Fig S11h is described as “relative contact frequency”, but shouldn’t this be a log fold difference (obs/exp) if it is indeed a ratio? Please clarify. Secondly, what is the origin of the clear blue cross centred along the boundary from leptotene to diplotene? If I am interpreting the figure correctly, this would mean that the boundary loci themselves have fewer contacts with all regions adjacent to them (on average) than expected by chance. How do the authors interpret this?

Moreover, I was unable to find (by searching) where Fig S11e-i are mentioned and discussed/described/interpreted in the text. Unless I am mistaken, please add appropriate text that describes and interprets these panels, or else remove them. It appears that some references to panels are incorrect and/or missing.

In Fig 2g. The title says pachytene/diplotene, but the legend states pachytene. Which is it?

Moreover, does pachytene/diplotene mean the average of pachytene+diplotene? Please clarify.

The analysis of compartment changes (Fig S15a-b) is much improved.

I thank the authors for clarifying the interpretation/description of Fig S15c. Nevertheless, I think this analysis would have been more powerful if it looked globally at interactions between compartment A/B rather than at this specific region of chromosome 6. (This suggested analysis is not necessary, but is just a comment on future potential analysis that might be more powerful/convincing.)

The analyses presented in the revised Fig 4 are interesting, but I would caution drawing too strong a conclusion from these rather weak effects present in A1 relative to A2-A3. Clearly, B behaves differently to A1-3 – a point that should be mentioned (I don’t believe that it currently is mentioned in the text).

Line 522. I suggest modifying the wording to: “...persist *over* a shorter fraction of *the* chromosome length”. I apologise for the grammatical mistake in my initial review.

Reviewer #2 (Remarks to the Author):

I recognize the effort the authors have made in addressing my comments. However, my initial concerns still remain. A big part of the results section is focussed on confirming previous studies. This includes description on HiC maps, TAD organization, CTCF/REC8 patterns, correlation between A-compartment and RNAseq data, correlation between A-compartment and DSBs, among others (see Wang et al. 2019; Patel et al. 2019; Alavattam et al. 2019; Vara et al. 2019).

As I stated in my initial report, pre-leptotene stage and Sertoli cells maps are new in the literature, but it is not clear how authors were able to differentiate among cell types during prophase I. For example, it is not clear what pre-leptotene stage means. In the M&M the authors state that 'Cells were categorized as preleptotene spermatocytes if they exhibited weak, diffused or punctate SYCP3 but no obvious stretches of SYCP3 signals'. But this is a pattern that can also be found in spermatogonia B.

Also, it is striking the similarity between Sertoli and metaphase II HiC heatmaps displayed in figure 1. It is known that HiC maps of chromosomes at metaphase stage display a very different pattern (see Naumova et al 2013, PMID: 24200812). This makes me wonder whether there was a clear distinction between cell types during the isolation process. In fact, Supplementary 3 shows high correlation values (0.90-0.92) between Sertoli and Metaphase II replicates. The same applies for the insulator profiles shown in Supplementary figure 4. There are also high correlation values (0.93-0.98) between leptotene, zygotene, pachytene and diplotene. Therefore, my initial concerns regarding the cell purification process still remain. I appreciate that the authors have added experiments of bulk RNA-seq, but I am afraid this doesn't confirm the purity of the isolated cells.

General Remarks

We would like to thank the editor and both reviewers for the helpful suggestions and comments. In this revised manuscript, we have made further changes to the texts and figures to address all of the specific questions raised by the reviewers. In addition, we performed a set of new analyses and revealed that the DSBs that would be resolved as crossovers (COs) or non-crossovers (NCOs) exhibited distinct chromatin interaction patterns during meiotic prophase I (Lines 296-328 in the revised manuscript), which to our knowledge has not been reported previously. Interestingly, we found that during preleptotene and leptotene stages, the CO-DSBs, but not the NCO-DSBs, exhibit a unique Hi-C interaction pattern that is reminiscent of TAD boundaries. The CO-DSBs also preferentially form long-range contacts with the neighboring regions. These findings suggest a link between the higher-order chromatin organization at the DSB sites and their future fates of whether to be resolved as COs or NCOs. The differences in chromatin organization patterns at these sites during preleptotene, leptotene, and zygotene stages also highlight the values of our stage-resolved Hi-C datasets for understanding the regulation of homolog interactions and recombination. We added a new Figure (Figure 3) to illustrate these new findings.

REVIEWER COMMENTS

Reviewer #1 (Remarks to the Author):

Comments on the revision: NCOMMS-20-36524A

Overall, the manuscript is much improved, and incorporates a wealth of data and detailed analyses. The author's should be commended for the depth of their study, and the thoughtful modifications made to the text and figures during revision. A number of small outstanding points/suggestions remain in the revision. It would be helpful if these could be addressed prior to publication.

We are grateful for Reviewer #1's constructive and detailed comments, which have led to significant improvement of our manuscript. We have incorporated new changes in texts and figures to address the points raised by the reviewer in our revised manuscript.

The explanation of the cell sorting and data presentation in Figure S1 (with the addition of panel b) is much improved. The new RNA-seq data in Fig S2 also show convincing differences between the sorted cell populations. Finally, the updated correlation matrix (at 500kb) better represents the differences (and similarities) between the sorted cell populations. This part of the manuscript is now greatly improved and much more informative.

Whilst Fig 1 (and especially 1f) is greatly improved, I remain skeptical of the comment about the "x-shape" and "coalescence of telomeres of different chromosomes" as presented in this figure. The authors may believe that this should

be the case, but the evidence (as presented in the figure) far from supports this, and certainly the pattern (as presented) is not “prominent”. All I observe is an increase in trans interactions in the centre of the plot between Chr1 and Chr 2, not an “X”. The much better analysis (and evidence) of this is presented in Figure 5, so I would suggest that the authors tone down this section (since, without evidence within the figure it is confusing to describe in this manner). The authors could optionally mention that this will be looked at in greater detail later on. It would also be helpful to indicate, on these zoom ins, which end of the chromosome is the centromere is (since these are telocentric chromosomes).

We agree with the reviewer that the trans-interaction patterns between Chr1 and Chr2 are not sufficient to illustrate the “coalescence of telomeres”. As such, we have toned down the statement and changed the original statement “Moreover, an ‘X’ shape inter-chromosomal interaction pattern becomes prominent at the zygotene stage (Fig. 1f), indicating the coalescence of telomeres of different chromosomes” to “The inter-chromosomal interaction patterns for different meiotic substages also exhibit discernible differences, with the interactions between the central regions of different chromosomes appearing strongest at the pachytene stage (Fig. 1f).” (Lines 164-167 in the revised manuscript) We have also indicated the centromere ends for both Chr1 and Chr2 on the zoom-ins in Fig. 1f.

Given that each chromosome is now presented separately in Fig S6-9 (which is very helpful and convincing!), it would be far more helpful for the reader (in terms of drawing comparisons), for the X-chromosome data currently on Figure S10, to be included as relevant panels in Figs S6-9. There is even a nice empty space for the X-chromosome panels in the bottom right of each figure. Importantly, overall (and certainly as presented!) I think it is hard to make the case that chromosome X is really any different than any one of the individual autosomes.

We thank the reviewer for this helpful suggestion on figure organization. We have now removed the original Figure S10 and moved the relevant panels for ChrX to Figs S6-9. We have also modified the statement regarding the progressive loop extension to “Such a pattern was reproducibly observed on each autosome, as well as the X chromosome (Supplementary Fig. 8,9).” (Lines 203-205 in the revised manuscript)

Fig S11h is described as “relative contact frequency”, but shouldn’t this be a log fold difference (obs/exp) if it is indeed a ratio? Please clarify. Secondly, what is the origin of the clear blue cross centred along the boundary from leptotene to diplotene? If I am interpreting the figure correctly, this would mean that the boundary loci themselves have fewer contacts with all regions adjacent to them (on average) than expected by chance. How do the authors interpret this?

Yes, the “relative contact frequency” in FigS11h (now FigS10h) stands for the obs/exp ratio, which we previously indicated in the figure legends. We have now also changed the labels on Figure axes to “obs/exp” to avoid confusion.

Regarding the blue crosses in Fig S11h. We found that some weak TAD boundaries (or local insulation minima to be exact) identified in spermatocytes were located at 10kb genomic bins that contained no uniquely mapped reads, thus appearing as blank lines on the heatmaps. However, these regions correspond to *bona fide* insulation minima as there were indeed lower levels of interactions occurring across these sites. A few examples are shown below in Reviewer Fig 1. Our interpretation for this phenomenon is that the insulation minima occurred at repetitive sequences at a higher frequency during meiosis than in somatic cells. We did not elaborate on this point in our manuscript as we think this phenomenon does not affect our conclusion that the TAD boundaries are weakened during the meiotic prophase I.

Reviewer Fig 1. TAD boundaries located at repetitive sequences

Hi-C observed interaction heatmaps (top row) and obs/exp heatmaps (bottom row) for five 1Mb genomic regions centered at repeats-containing TAD boundaries identified in the pachytene stage. Empty crosses are observed at the center of these regions due to the lack of uniquely mapped reads.

Moreover, I was unable to find (by searching) where Fig S11e-i are mentioned and discussed/described/interpreted in the text. Unless I am mistaken, please add appropriate text that describes and interprets these panels, or else remove them. It appears that some references to panels are incorrect and/or missing.

We have now correctly referenced all Figure S11 panels (now Figure S10 after reorganization. Lines 218 to 227 in the revised manuscript). We have also checked and corrected the figure references throughout the manuscript.

In Fig 2g. The title says pachytene/diplotene, but the legend states pachytene. Which is it?

Moreover, does pachytene/diplotene mean the average of pachytene+diplotene? Please clarify.

We thank the reviewer for pointing this out. The “pachytene/diplotene” means that these ChIP experiments were performed on the mixed-population of pachytene and diplotene spermatocytes that were isolated using less stringent FACS parameters. We have now corrected the figure legend for Fig 2g to “pachytene/diplotene”. We have also edited the text and the relevant material and methods sections to clarify this point

(Lines 245 to 246, 845 to 850, 1027 to 1029 in the revised manuscript).

The analysis of compartment changes (Fig S15a-b) is much improved.

I thank the authors for clarifying the interpretation/description of Fig S15c. Nevertheless, I think this analysis would have been more powerful if it looked globally at interactions between compartment A/B rather than at this specific region of chromosome 6. (This suggested analysis is not necessary, but is just a comment on future potential analysis that might be more powerful/convincing.)

We appreciated the reviewer's suggestion and will perform these analyses in our future studies.

The analyses presented in the revised Fig 4 are interesting, but I would caution drawing to strong a conclusion from these rather weak effects present in A1 relative to A2-A3. Clearly, B behaves differently to A1-3 – a point that should be mentioned (I don't believe that it currently is mentioned in the text).

We appreciate the reviewer's comment regarding this section. In the revised manuscript, we now describe the differences in CO/ DSB ratios between the A and B compartment regions before getting onto the differences between A1 and A2/3. We have also toned down the conclusion for this section to “these results suggest that the enrichment of short chromatin loop sizes is associated with a higher conversion rate of DSBs to COs” (Lines 420 to 437 in the revised manuscript). We have modified Figure 4 (now Figure 5 in the revised manuscript) and added the P values for comparisons between A1-3 and B.

Line 522. I suggest modifying the wording to: “...persist *over* a shorter fraction of *the* chromosome length”. I apologise for the grammatical mistake in my initial review.

We have modified the wording as the reviewer suggested (Lines 537 to 540 in the revised manuscript).

Reviewer #2 (Remarks to the Author):

I recognize the effort the authors have made in addressing my comments. However, my initial concerns still remain. A big part of the results section is focused on confirming previous studies. This includes description on HiC maps, TAD organization, CTCF/REC8 patterns, correlation between A-compartment and RNAseq data, correlation between A-compartment and DSBs, among others (see Wang et al. 2019; Patel et al. 2019; Alavattam et al. 2019; Vara et al. 2019).

We appreciate reviewer #2's comments, which have prompted us to conduct more thorough data analyses and a more stringent quality assessment of our data in our revised manuscript.

One important distinction between our study and the several previously published Hi-C studies is that the previous studies mainly focused on how the changes in meiotic chromosome organization may influence or coordinate the transcription programs during gametogenesis. In contrast, our study focuses on the chromosome organization features that correlate with or may influence homolog alignment and recombination. Taking advantage of our stage-resolved Hi-C datasets, we performed a series of detailed analyses that revealed new Hi-C interaction patterns and insights, which we emphasized in each of our seven main Figures. In our responses below, we would like to further outline the novel findings presented by our study.

- 1. First and foremost, we would like to clarify that in the current study, we not only generated the Hi-C maps for preleptotene stage and Sertoli cells, but also the first Hi-C maps for leptotene and diplotene stages (The Vara *et al.* 2019 study profiled the chromosome architecture in the mixed populations of leptotene/zygotene and pachytene/diplotene spermatocytes, but was not able to resolve the individual stages.). Thus, our efforts have generated a complete, stage-resolved Hi-C dataset for evaluating the temporal progression of chromosome organization throughout the entire meiotic prophase I. In Figure1, we demonstrated the feasibility to purify cells at all meiotic prophase I substages at desirable quantities for Hi-C analyses. We anticipate such a purification strategy would benefit future studies in the meiosis field.**
- 2. In Figure2, we focused on describing how the meiotic chromosome loops behave differently from the well-studied, CTCF-anchored chromatin loops in interphase. Our analysis suggests that tethering chromatin loops to the meiotic chromosome axis does not create strong barriers for loop extension, which has not been directly assessed in mammalian cells before our study.**
- 3. In the newly generated Figure3, we evaluated the higher-order chromatin structure around different classes of DSBs throughout meiotic prophase I. These new analyses revealed that the CO-favored and the CO-disfavored**

DSBs exhibit distinct Hi-C interaction patterns, which to our knowledge have not been reported previously.

- 4. In Figure 4, we demonstrated that the A/B compartment exhibited different sizes of chromatin loops during leptotene and zygotene stages, with the most pronounced differences observed in leptotene. These analyses provide the first molecular-level characterization of chromatin loop size variations across the meiotic chromosomes.**
- 5. In Figure 5, we further explored the functional significances of the chromatin structure variations across the chromosome in homolog alignment and recombination. We showed that the variations in loop size correlate with the extent of homolog interactions, as well as the probability of CO occurrence. We note that the Patel *et al.* 2019 study showed that the distribution of DSB hotspots is biased toward the active A compartment, and attributed this bias to the enrichment of PRDM9 binding sites and H3K4me3 peaks in the A compartment. However, we show that the differences in chromatin loop sizes/ compaction rate correlate with the probabilities of converting DSBs into COs. These findings suggest a functional link between higher-order chromosome organization and CO regulation across the genome.**
- 6. In Figures 6 and 7, we characterized the patterns of the interchromosomal interactions between subtelomeric regions that persist over a significant fraction of chromosome length. Such a phenomenon has not been explicitly assessed before. Importantly, we also showed that the alignment of subtelomeric regions and the association of the telomeres can be decoupled using a LINC-mutant mouse model that we recently developed (Chen *et al.* Nature Communications 2021, PMID: 3403999), thus revealing a novel function of the mechanical-transmitting LINC complex during meiosis progression.**

While the above being said, we also felt obligated to characterize some previously reported meiotic chromosome features, such as meiotic chromatin loops sizes and TAD boundary strength, at a higher temporal resolution, as we anticipate this information would be valuable to future studies in meiosis field. Most of those analyses were presented in our 19 supplementary figures and were not the focus of our manuscript.

As previously suggested by Patel *et al.* at the end of their 2019 Nature Structural and Molecular Biology paper, “Newly developed methods for the synchronization and purification of mouse spermatocytes, and their analysis by Hi-C, can provide a new window into the organization and function of meiotic chromosomes. These advances will be critical to increasing our understanding of the roles of structural proteins, including cohesin and chromosome axis components, the interplay between chromosome organization and transcription, and homolog interactions during recombination and synapsis.” We believe that the data and analyses presented in our study are an important step towards achieving this goal.

As I stated in my initial report, pre-leptotene stage and Sertoli cells maps are new in the literature, but it is not clear how authors were able to differentiate among cell types during prophase I. For example, it is not clear what pre-leptotene stage means. In the M&M the authors state that ‘Cells were categorized as preleptotene spermatocytes if they exhibited weak, diffused or punctate SYCP3 but no obvious stretches of SYCP3 signals’. But this is a pattern that can also be found in spermatogonia B.

Regarding the general feasibility to differentiate among cell types during prophase I, we would like to point out that it has been well documented that different spermatocyte cell types occupy distinct areas on the Hoechst profiles from FACS (PMC4246648). For instance, the spermatogonia, leptotene/zygotene, and pachytene/diplotene cells appear as separate, dense clusters of cells on the Hoechst profile. Furthermore, the criteria used to distinguish different spermatocyte cell types based on SYCP3 and H2AX staining were well established. In fact, in the original paper describing the FACS-based approach for isolating spermatocytes, six different cell types were successfully isolated from a single experiment from adult mice (2-5 months old) at >75% purity (see the table from PMC4246648 below). However, further improving the purity of each spermatocyte type requires a significant amount of work. In our study, we isolated spermatocytes of different meiotic substages at greater or close to 90% purity by using mice of proper ages for different cell types, performing cell sorting with stringent gating parameters, and meticulously assessing the purity of small batches of isolated cells.

COLLECTED / OBSERVED	SPG	PREL	L	Z	P	D
Spermatogonia (Spg)	80-91%	5-10%	3%	0	0	0
Preleptotene (PreL)	5-10%	75-92%	5-10%	0	0	0
Leptotene (L)	0	5-10%	60%-80%	5-10%	0	0
Zygotene (Z)	0	0	10-15%	75-90%	3%	2%
Pachytene (P)	0	0	3%	10-15%	81-95%	4-10%
Diplotene (D)	0	0	0	0	5-7%	82-95%
Other	4-5%	1-2%	1%	1%	1-2%	2-3%

Reviewer Table 1. Percent purity quantification based on immunofluorescence analysis after cell sorting from a previous study

The table is taken from PMC4246648. In this study, six different cell types were purified in the same cell sorting experiment. The cell purity for each cell type was estimated from 8 experiments.

By definition, preleptotene spermatocytes represent the spermatocytes that have completed the last mitotic division and have not entered meiosis yet. Previous FACS studies have shown that the preleptotene cells are located at the elongated regions bridging the spermatogonia region and the leptotene/zygotene region on the Hoechst profiles, thus can be separated from the main spermatogonia populations using proper gating windows. In our study, we used the SYCP3 staining to further verify the

identity of the preleptotene spermatocytes and distinguish them from spermatogonia cells, as previous literature has demonstrated that the spermatogonia cells should exhibit no detectable SYCP3 staining (PMID 23318132, 24664803, 31358751, 32032549). Although the preleptotene cells purified in our study exhibited a variety of SYCP3 staining patterns, the SYCP3 signals were readily visible in >90% of purified cells (please see Reviewer Figure 2 for representative fluorescent images of a few microscopic fields). Moreover, the P(s) curves and chromatin loop sizes (Figure 2) derived from the preleptotene Hi-C data exhibit an intermediate shape between those for spermatogonia and leptotene spermatocytes, further supporting that these cells represent the transition stage between proliferating spermatogonia and meiotic spermatocytes.

Reviewer Fig 2. Isolated preleptotene spermatocytes exhibit visible SYCP3 staining

Representative immunofluorescence (IF) images of isolated preleptotene spermatocytes stained for meiosis markers SYCP3 (green), H2Ax (red), and DAPI (blue). SYCP3 signals are visible in nearly all isolated preleptotene cells. Scale bars, 50 μ m.

Also, it is striking the similarity between Sertoli and metaphase II HiC heatmaps displayed in figure 1. It is known that HiC maps of chromosomes at metaphase stage display a very different pattern (see Naumova et al 2013, PMID: 24200812). This makes me wonder whether there was a clear distinction between cell types during the isolation process. In fact, Supplementary 3 shows high correlation values (0.90-0.92) between Sertoli and Metaphase II replicates. The same applies for the insulator profiles shown in Supplementary figure 4. There are also high correlation values (0.93-0.98) between leptotene, zygotene, pachytene and diplotene. Therefore, my initial concerns regarding the cell purification process still remain. I appreciate that the authors have added experiments of bulk RNA-seq, but I am afraid this doesn't confirm the purity of the isolated cells.

We appreciate the reviewer for raising the point regarding the "Metaphase II" spermatocytes. The spermatocytes that we previously named as "Metaphase II" were the 2C cells localized at the bottom right corner on the Hoechst profiles. Upon further checking the references, we now realize that these 2C cells actually represent "Meiosis II" spermatocytes, which should not only include the Metaphase II cells but also the interphase II cells. Therefore, it now becomes understandable why those "Metaphase II" cells exhibit higher similarity in TAD and compartment patterns, as well as higher correlation coefficients, with the Sertoli cells. We apologize for the misinterpretation of the identity of those cells and have now corrected the "Metaphase II" to "Meiosis II" cells throughout the manuscript.

In response to the reviewer's concerns that the high correlation values between Hi-C data of different meiosis stages may indicate unsuccessful purification procedures, we would like to emphasize that we used the distinct SYCP3 and H2AX staining patterns as well as the chromosome morphology as the ultimate criteria for assessing the cell identity. For instance, the pachytene and diplotene cells could be easily discriminated based on whether the chromosome axes were fully synapsed or partially separated. And the zygotene and the pachytene cells could be unambiguously distinguished based on the appearance of the X-Y sex body. These well-established criteria for assessing the stages of spermatocytes have been routinely used in our labs and have been demonstrated in several of our recent publications (Long *et al.*, Nature Structural and Molecular Biology 2017, PMID: 29083416; Wang *et al.*, Nature Communications 2019, PMID: 30718482; Chen *et al.*, Nature Communications 2021, PMID: 3403999). We have high confidence in the validity and accuracy of our purity assessment on the isolated spermatocytes.

Since the chromosomes at different meiotic substages have all adopt a linearly arranged loop array conformation, high correlation values between the Hi-C datasets would be expected. Despite the high correlation coefficients, our Hi-C datasets did exhibit stage-dependent differences in a variety of chromatin organization features, such as the progressive increases in chromatin loop size (Fig2), the unique chromatin interaction patterns around the DSBs in preleptotene and leptotene (Fig3), the clustering and alignment of chromosome ends in leptotene and zygotene (Fig6), etc.

All of the evidence above suggests that our purification strategy successfully resolved the spermatocytes of different meiotic substages.

We also evaluated the correlation coefficients between the Hi-C datasets from the two previous studies that performed Hi-C analyses on more than one meiotic stage (Patel *et al.* 2019 and Vara *et al.* 2019). In the Vara *et al.* 2019 Cell Reports study, the leptotene/ zygotene and the pachytene/ diplotene mixed-populations were isolated using a highly similar approach to our study but with less stringent gating conditions. The correlation coefficient between the leptotene/ zygotene and the pachytene/ diplotene Hi-C heatmaps from their study was 0.91. In comparison, the correlation coefficients between leptotene and pachytene/ diplotene in our study are 0.83-0.86. And those between zygotene and pachytene/ diplotene in our study are 0.90-0.96. Thus, the extent of correlation between Hi-C datasets in our study would be consistent with the Vara *et al.* study if the leptotene/ zygotene and the pachytene/ diplotene stages were mixed.

However, the correlation coefficients between the zygotene and pachytene Hi-C heatmaps from the Patel *et al.* 2019 Nature Structural and Molecular Biology study were 0.74-0.86, much lower than the values from both our and Vara's studies. One possible explanation for such a discrepancy is that the Patel *et al.* study purified both the zygotene and pachytene spermatocytes using a spermatogenesis synchronization strategy. Such a purification strategy could yield more homogenous zygotene cell populations biasing towards early- or mid-zygotene, or pachytene cell populations biasing towards late-pachytene, thereby giving rise to greater differences between the Hi-C datasets.

Reviewer Fig 3. Correlation between Hi-C datasets of different meiotic stages

Heatmaps show correlations between pairwise combinations of Hi-C datasets generated in Vara *et al.* 2019 study (a), Patel *et al.* 2019 study (b), and this study (c). Pearson correlation coefficients were calculated using balanced chromatin interactions binned at 500kb and the HiCRep package in R as in Supplementary Figure 3 in our manuscript.

REVIEWERS' COMMENTS

Reviewer #1 (Remarks to the Author):

I thank the reviewers for their detailed response to my prior comments. I have two outstanding (minor queries):

1. The authors mention that the centromere ends are marked on Fig 1f. However, I cannot see this addition.
2. Fig S11h. The authors appear to suggest that the blue cross arises from bins that have few reads (due to multimapping). If this is indeed the case, then I think this explanation should be clearly stated in the legend (i.e. that it is an artefact of the data aggregation including numerous loci with zero signal.)

We would like to thank the editor and the reviewers again for the helpful suggestions and comments throughout the review process. In this revised manuscript, we have made further changes to the texts and figures to address the two minor issues raised by the Reviewer #1.

REVIEWERS' COMMENTS

Reviewer #1 (Remarks to the Author):

I thank the reviewers for their detailed response to my prior comments. I have two outstanding (minor queries):

1. The authors mention that the centromere ends are marked on Fig 1f. However, I cannot see this addition.

We apologize for the omission of the labels in the previous submission. We have now added the label “cen” to indicate the centromere ends for both chr1 and chr2 in Fig 1f.

2. Fig S11h. The authors appear to suggest that the blue cross arises from bins that have few reads (due to multimapping). If this is indeed the case, then I think this explanation should be clearly stated in the legend (i.e. that it is an artefact of the data aggregation including numerous loci with zero signal.)

We have added the following sentence “The blue crosses observed at the center of the heatmaps arise from the large number of TAD boundaries located at genomic bins lacking uniquely mapped reads.” To the figure legend of Fig S11h to explain the cause of the blue crosses.